# Resilient Constrained Learning

**Ignacio Hounie**
University of Pennsylvania

**Alejandro Ribeiro**
University of Pennsylvania

**Luiz F. O. Chamon**
University of Stuttgart

## Abstract

When deploying machine learning solutions, they must satisfy multiple requirements beyond accuracy, such as fairness, robustness, or safety. These requirements are imposed during training either implicitly, using penalties, or explicitly, using constrained optimization methods based on Lagrangian duality. Either way, specifying requirements is hindered by the presence of compromises and limited prior knowledge about the data. Furthermore, their impact on performance can often only be evaluated by actually solving the learning problem. This paper presents a constrained learning approach that adapts the requirements while simultaneously solving the learning task. To do so, it relaxes the learning constraints in a way that contemplates how much they affect the task at hand by balancing the performance gains obtained from the relaxation against a user-defined cost of that relaxation. We call this approach *resilient constrained learning* after the term used to describe ecological systems that adapt to disruptions by modifying their operation. We show conditions under which this balance can be achieved and introduce a practical algorithm to compute it, for which we derive approximation and generalization guarantees. We showcase the advantages of this resilient learning method in image classification tasks involving multiple potential invariances and in heterogeneous federated learning.

## 1 Introduction

Requirements are integral to engineering and of growing interest in machine learning (ML) [1]. This growing interest is evident in, e.g., the advancement towards designing ML systems that are fair [2], robust [3], and safe [4], as well as numerous applications in which we want to attain good performance with respect to more than one metric [5–24]. Two concrete applications that we will use as examples (Section 5) are *heterogeneous federated learning*, where each client realizes a different loss due to distribution shifts (as in, e.g., [18–20]) and *invariant learning*, where we seek to achieve good performance even after the data has undergone a variety of transformations (as in, e.g., [15, 21–23]).

The goal in these settings is to strike a compromise between some top-line objective metric and the requirements. To this end, an established approach is to combine the top-line and requirement violation metrics in a single training loss. This leads to *penalty methods* that are ubiquitous in ML, as attested by, e.g., fairness [25–28] and robustness [29, 30] applications. Another approach to balance objective and requirements is formulating and solving constrained learning problems. Though less typical in ML practice, they are not uncommon [5–18, 21, 24]. In particular, they have also been used in fair [31–34] and robust [12, 35] learning, to proceed with a common set of applications. It is worth noting that penalty and constrained methods are not unrelated. They are in fact equivalent in convex optimization, in the sense that every constrained problem has an equivalent penalty-based formulation. A similar result holds in non-convex ML settings for sufficiently expressive parametrizations [7, 8].

In either case, the compromise between objective and different requirements are specified by (hyper)parameters, be they penalty coefficients or constraint levels (Section 2). Finding penalties or constraints specifications that yield reasonable trade-offs is particularly difficult in ML, which often

involves statistical requirements, such as fairness and robustness, that have intricate dependencies with the model and unknown data distributions. Case in point, consider invariant learning for image classification [23]. While we know that invariance to rotations and translations is desirable, we do not know *how much* invariance to these transformations is beneficial. This depends on the level of invariance of the data distribution, its prevalence in the dataset, and the capability of the model to represent invariant functions. The standard solution to this problem involves time consuming and computationally expensive hyperparameter searches.

This paper addresses this issue by automating the specification of constraint levels during training. To do so, it begins by interpreting constraints as nominal specifications that can be relaxed to find a better compromise between objective and requirements (Section 3). We call this approach *resilient constrained learning* after the term used to describe ecological systems that adapt to disruptions by modifying their operation [36, 37]. Our first contribution is the following insight:

(C1)  We relax constraints according to their relative difficulty, which we define as the sensitivity of the objective to perturbations of the constraint (Section 3.1).

That difficult constraints should be relaxed more is a natural choice. The value of (C1) is in defining what is a difficult constraint. We then seek constraint levels such that the objective loss is relatively insensitive to changes in those levels. This relative insensitivity incorporates a user-defined cost that establishes a price for relaxing nominal specifications.

The learning problem implied by (C1) seems challenging. Our next contribution is to show that it is not:

(C2)  We use duality and perturbation theory to present reformulations of the resilient learning problem from (C1) (Section 3.2) that lead to a practical resilient learning algorithm (Section 4) for which we derive statistical approximation bounds (Thm. 1).

Our final contribution is the experimental evaluation of the resilient learning algorithm:

(C3)  We evaluate resilient formulations of federated learning and invariant learning (Section 5).

Our experiments show that (C1)–(C2) effectively relaxes constraints according to their *difficulty*, leading to solutions that are *less sensitive* to the requirement specifications. It illustrates how resilient learning constitutes an interpretable and flexible approach to designing requirements while contemplating performance trade-offs.

## 2   Learning with Constraints

Let $\mathcal{D}_0$ be a distribution over data pairs $(\mathbf{x}, y)$ composed of the feature vector $\mathbf{x} \in \mathcal{X} \subset \mathbb{R}^d$ and the corresponding output $y \in \mathcal{Y} \subset \mathbb{R}$. Let $f_\theta : \mathcal{X} \to \mathbb{R}^k$ be the function associated with parameters $\theta \in \Theta \subset \mathbb{R}^p$ and $\ell_0 : \mathbb{R}^k \times \mathcal{Y} \to [-B, B]$ be the loss that evaluates the fitness of the estimate $f_\theta(\mathbf{x})$ relative to $y$. Let $\mathcal{F}_\theta = \{f_\theta \mid \theta \in \Theta\}$ be the hypothesis class induced by these functions. Different from traditional (unconstrained) learning, we do not seek $f_\theta$ that simply minimizes $\mathbb{E}[\ell_0(\phi(\mathbf{x}), y)]$, but also account for its expected value with respect to additional losses $\ell_i : \mathbb{R}^k \times \mathcal{Y} \to [-B, B]$ and distributions $\mathcal{D}_i, i = 1, \dots, m$. These losses/distributions typically encode statistical requirements, such as robustness (where $\mathcal{D}_i$ denote distribution shifts or adversarial perturbations) and fairness (where $\mathcal{D}_i$ are conditional distributions of protected subgroups).

Explicitly, the constrained statistical learning (CSL) problem is defined as

$$\begin{aligned} \mathsf{P}^\star = \min_{\theta \in \Theta} \quad & \mathbb{E}_{(\mathbf{x}, y) \sim \mathcal{D}_0} \Big[ \ell_0 \big( f_\theta(\mathbf{x}), y \big) \Big] \\ \text{subject to} \quad & \mathbb{E}_{(\mathbf{x}, y) \sim \mathcal{D}_i} \Big[ \ell_i \big( f_\theta(\mathbf{x}), y \big) \Big] \le 0, \quad i = 1, \dots, m. \end{aligned} \tag{P}$$

Without loss of generality, we stipulate the nominal constraint specification to be zero. Other values can be achieved by offsetting $\ell_i$, i.e., $\mathbb{E}\big[\tilde{\ell}(f_\theta(\mathbf{x}), y)\big] \le c$ is obtained using $\ell_i(\cdot) = \tilde{\ell}(\cdot) - c$ in (P). We also let $\mathsf{P}^\star$ take values on the extended real line $\mathbb{R} \cup \{\infty\}$ by defining $\mathsf{P}^\star = \infty$ whenever (P) is infeasible, i.e., whenever for all $\theta \in \Theta$ there exists $i$ such that $\mathbb{E}_{\mathcal{D}_i}\big[\ell_i(f_\theta(\mathbf{x}), y)\big] > 0$.

A challenge in formulating meaningful CSL problems lies in specifying the constraints, i.e., the $\ell_i$. Indeed, while a solution $f_{\theta^\star}$ always exists for unconstrained learning [$m = 0$ in (P)], there may be no $\theta$ that satisfies the constraints in (P). This issue is exacerbated when solving the problem using data: even arbitrarily good approximations of the expectations in (P) may introduce errors that hinder the estimation of $P^\star$ (see Appendix A for a concrete example). In practice, landing on feasible requirements may require some constraints to be relaxed relative to their initial specification. Then, in lieu of (P), we would use the relaxed problem

$$
\begin{aligned}
P^\star(\mathbf{u}) = \min_{\theta \in \Theta} \quad & \mathbb{E}_{(\mathbf{x},y) \sim \mathcal{D}_0}\Big[\ell_0\big(f_\theta(\mathbf{x}), y\big)\Big] \\
\text{subject to} \quad & \mathbb{E}_{(\mathbf{x},y) \sim \mathcal{D}_i}\Big[\ell_i\big(f_\theta(\mathbf{x}), y\big)\Big] \le u_i, \quad i = 1, \ldots, m,
\end{aligned}
\tag{$P_\mathbf{u}$}
$$

where $\mathbf{u} \in \mathbb{R}_+^m$ collects the relaxations $u_i \ge 0$. The value $P^\star(\mathbf{u})$ is known as the *perturbation function* of (P) since it describes the effect of the relaxation $\mathbf{u}$ on the optimal value. Given $P^\star(\mathbf{0}) = P^\star$ and ($P_\mathbf{0}$) is equivalent to (P), this abuse of notation should not lead to confusion.

It is ready that $P^\star(\mathbf{u})$ is a componentwise non-increasing function, i.e., that for comparable arguments $v_i \le w_i$ for all $i$ (denoted $\mathbf{v} \preceq \mathbf{w}$), it holds that $P^\star(\mathbf{v}) \ge P^\star(\mathbf{w})$. However, large relaxations $\mathbf{u}$ drive ($P_\mathbf{u}$) away from (P), the learning problem of interest. Thus, relaxing (P) too much can be as detrimental as relaxing it too little (see example in Appendix A). The goal of this paper is to exploit properties of $P^\star(\mathbf{u})$ together with a relaxation cost to strike a balance between these conflicting objectives. We call this balance a *resilient* version of (P).

## 3 Resilient Constrained Learning

We formulate resilient constrained learning using a functional form of ($P_\mathbf{u}$). Explicitly, consider a *convex* function class $\mathcal{F} \supseteq \mathcal{F}_\theta$ and define the relaxed functional problem

$$
\begin{aligned}
\tilde{P}^\star(\mathbf{u}) = \min_{\phi \in \mathcal{F}} \quad & \mathbb{E}_{(\mathbf{x},y) \sim \mathcal{D}_0}\Big[\ell_0\big(\phi(\mathbf{x}), y\big)\Big] \\
\text{subject to} \quad & \mathbb{E}_{(\mathbf{x},y) \sim \mathcal{D}_i}\Big[\ell_i\big(\phi(\mathbf{x}), y\big)\Big] \le u_i, \quad i = 1, \ldots, m.
\end{aligned}
\tag{$\tilde{P}_\mathbf{u}$}
$$

The difference between ($P_\mathbf{u}$) and ($\tilde{P}_\mathbf{u}$) is that the latter does not rely on a parametric model. Instead, its solutions take values on the convex space of functions $\mathcal{F}$. Still, if $\mathcal{F}_\theta$ is a sufficiently rich parameterization of $\mathcal{F}$ (see Section 4 for details), then $P^\star(\mathbf{u})$ and $\tilde{P}^\star(\mathbf{u})$ are close [38, Sec. 3]. Throughout the rest of the paper, we use $\sim$ to identify these functional problems.

The advantage of working with ($\tilde{P}_\mathbf{u}$) is that under mild conditions the perturbation function $\tilde{P}^\star$ is convex. This holds readily if the losses $\ell_i$ are convex (e.g., quadratic or cross-entropy) [39, chap. 5]. However, the perturbation function is also convex for a variety of non-convex programs, e.g., ($\tilde{P}_\mathbf{u}$) with non-convex losses, non-atomic $\mathcal{D}_i$, and decomposable $\mathcal{F}$ (see Appendix B). For conciseness, we encapsulate this hypothesis in an assumption.

**Assumption 1 .** The perturbation function $\tilde{P}^\star(\mathbf{u})$ is a convex function of the relaxation $\mathbf{u} \in \mathbb{R}_+^m$.

A consequence of Assumption 1 is that the perturbation function $\tilde{P}^\star$ has a non-empty subdifferential at every point. Explicitly, let its subdifferential $\partial \tilde{P}^\star(\mathbf{u}^o)$ at $\mathbf{u}^o \in \mathbb{R}_+^m$ be defined as the set of vectors describing supporting hyperplanes at $\mathbf{u}^o$ of the epigraph of $\tilde{P}^\star$, i.e.,

$$
\partial \tilde{P}^\star(\mathbf{u}^o) = \Big\{ \mathbf{p} \in \mathbb{R}_+^m \mid \tilde{P}^\star(\mathbf{v}) \ge \tilde{P}^\star(\mathbf{u}_o) + \mathbf{p}^T(\mathbf{v} - \mathbf{u}_o), \text{ for all } \mathbf{v} \in \mathbb{R}_+^m \Big\}.
\tag{1}
$$

The elements of $\partial \tilde{P}^\star(\mathbf{u}^o)$ are called *subgradients* of $\tilde{P}^\star$ at $\mathbf{u}^o$. If $\tilde{P}^\star$ is differentiable at $\mathbf{u}^o$, then it has a single subgradient that is equal to its gradient, i.e., $\partial \tilde{P}^\star(\mathbf{u}^o) = \{\nabla \tilde{P}^\star(\mathbf{u}^o)\}$. In general, however, the subdifferential is a non-singleton set [40]. Further notice that since $\tilde{P}^\star(\mathbf{u})$ is componentwise nonpositive the subgradients are componentwise negative, $p_\mathbf{u} \preceq \mathbf{0}$.

Next, we use this property to formalize resilient constrained learning as a compromise between reducing $\tilde{P}^\star(\mathbf{u})$ by increasing $\mathbf{u}$ and staying close to the original problem ($\tilde{P}_\mathbf{0}$).

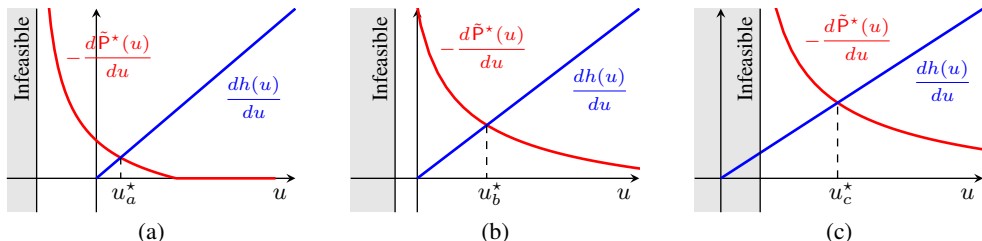

Figure 1: Resilient equilibrium from Def. 1 for $h(u) = u^2/2$. The shaded area indicates infeasible specifications: (a) nominal specification ($u = 0$) is feasible and easy to satisfy; (b) nominal specification is feasible but difficult to satisfy (close to infeasible); (c) nominal specification is infeasible.

## 3.1 Resilient Equilibrium

Consider the effect of increasing the value of a specific relaxation $u_i$ in ($\tilde{P}_{\mathbf{u}}$) while keeping the rest unchanged. The solution of ($\tilde{P}_{\mathbf{u}'}$) is allowed to suffer higher losses on the constraints ($\ell_i$ for $i \geq 1$), which may lead to a smaller objective loss ($\ell_0$). Hence, while larger relaxations are detrimental because they violate more the requirements, they are also beneficial because they reduce the objective loss. To balance these conflicting outcomes of constraint relaxations, we introduce a function $h$ to capture their costs. Then, since relaxing both increases the costs and decreases the objective value, we conceptualize resilience as an equilibrium between these variations.

> **Definition 1 (Resilient Equilibrium).** Let $h : \mathbb{R}_+^m \rightarrow \mathbb{R}_+$ be a convex, differentiable, normalized (i.e., $h(\mathbf{0}) = 0$), and componentwise increasing (i.e., $h(\mathbf{v}) < h(\mathbf{w})$ for $\mathbf{v} \prec \mathbf{w}$) function. A resilient equilibrium of ($\tilde{P}_{\mathbf{u}}$) is a relaxation $\mathbf{u}^\star$ satisfying
>
> $$\nabla h(\mathbf{u}^\star) \in -\partial \tilde{P}^\star(\mathbf{u}^\star). \tag{2}$$

The resilient constrained learning problem amounts to solving ($\tilde{P}_{\mathbf{u}^\star}$), i.e., solving ($\tilde{P}_{\mathbf{u}}$) for a relaxation $\mathbf{u}^\star$ that satisfies (2). We call this equilibrium *resilient* because it describes how far ($\tilde{P}_0$) can be relaxed before we start seeing diminishing returns. Indeed, $\mathbf{u}^\star$ from (2) is such that relaxing by an additional $\boldsymbol{\epsilon} \succ \mathbf{0}$ would incur in a relaxation cost at least $\nabla h(\mathbf{u}^\star)^T \boldsymbol{\epsilon}$ larger, whereas tightening to $\mathbf{u}^\star - \boldsymbol{\epsilon}$ would incur in an optimal value increase of at least the same $\nabla h(\mathbf{u}^\star)^T \boldsymbol{\epsilon}$.

Notice that resilient constrained learning is defined in terms of *sensitivity*. Indeed, the resilient equilibrium in Def. (1) specifies a learning task that is as sensitive to changes in its requirements as it is sensitive to changes in the relaxation cost. This has the marked advantage of being invariant to constant translations of $\ell_0$, as is also the case for solutions of ($\tilde{P}_{\mathbf{u}}$). Sensitivity also measures the difficulty of satisfying a constraint, since $\partial \tilde{P}^\star(\mathbf{u})$ quantifies the impact of each constraint specification on the objective loss. Hence, the equilibrium in (2) has the desirable characteristic of affecting stringent requirements more.

Two important properties of the equilibrium in Def. 1 are summarized next (proofs are provided in appendices D.1 and D.2).

**Proposition 1.** *Under Ass. 1, the resilient equilibrium* (2) *exists. If $h$ is strictly convex, it is unique.*

**Proposition 2.** *Let $\mathbf{v}, \mathbf{w} \in \mathbb{R}_+^m$ be such that $[\mathbf{v}]_i = [\mathbf{w}]_i$, for $i \neq j$, and $[\mathbf{v}]_j < [\mathbf{w}]_j$. Under Ass. 1, (i) $[\nabla h(\mathbf{v})]_j \leq [\nabla h(\mathbf{w})]_j$ and (ii) $[-\mathbf{p}_v]_j \geq [-\mathbf{p}_w]_j$ for all $\mathbf{p}_v \in \partial \tilde{P}^\star(\mathbf{v})$ and $\mathbf{p}_w \in \partial \tilde{P}^\star(\mathbf{w})$.*

Prop. 1 shows that the equilibrium in (2) is well-posed. Prop. 2 states that, all things being equal, relaxing the $j$-th constraint increases the sensitivity of the cost $h$ to it, while simultaneously decreasing its effect on the objective value $\tilde{P}^\star$. To illustrate these points better, Fig. 1 considers prototypical learning problems with a single constraint [$m = 1$ in ($\tilde{P}_{\mathbf{u}}$)], differentiable $\tilde{P}^\star(u)$, and relaxation cost $h(u) = u^2/2$. According to (2), the resilient relaxations are obtained at $h'(u^\star) = u^\star = \tilde{P}^{\star'}(u^\star)$, where we let $g'(u) = dg(u)/du$ denote the derivative of the function $g$. As per Prop. 2, $h'$ is increasing and $-\tilde{P}^{\star'}$ is decreasing. Further observe that the sensitivity $-\tilde{P}^{\star'}(u)$ diverges as $u$ approaches the value that makes the problem infeasible and vanishes as the constraint is relaxed. These two curves must therefore intersect, claimed by Prop. 1.

The illustrations in Fig. 1 represent progressively more sensitive/difficult problems. In Fig. 1a, the nominal problem ($u = 0$) is easy to solve (small $\tilde{\mathsf{P}}^{\star\prime}(0)$), making the resilient equilibrium $u_a^* \approx 0$. The original problem and the relaxed problem are essentially equivalent. In Fig. 1b, the nominal problem is difficult to solve (large $\tilde{\mathsf{P}}^{\star\prime}(0)$), inducing a significant change in the constraint and objective losses. In Fig. 1c, the nominal problem is unsolvable, but the resilient relaxation recovers a feasible problem.

Having motivated the resilient compromise in Def. 1 and proven that it is well-posed, we proceed to obtain equivalent formulations that show it is also computationally tractable. These formulations are used to show traditional learning tasks that can be seen as resilient learning problems (Sec. 3.3), before deriving a practical algorithm to tackle the resilient learning problem ($\tilde{\mathsf{P}}_{\mathbf{u}^\star}$) (Sec. 4).

## 3.2 Equivalent Formulations

While we have shown that the resilient relaxation from Def. 1 exists (Prop. 1), the equilibrium in (2) does not provide a straightforward way to compute it. In this section, we show two more computationally amenable reformulations of Def. 1 by relating $\mathbf{u}^\star$ to the Lagrange multipliers of ($\tilde{\mathsf{P}}_{\mathbf{u}}$) and to the solution of a related optimization problem.

Let $\boldsymbol{\lambda} \in \mathbb{R}_+^m$ collect multipliers $\lambda_i$ associated to the $i$-th constraint of ($\tilde{\mathsf{P}}_{\mathbf{u}}$) and define the Lagrangian

$$\mathcal{L}(\phi, \boldsymbol{\lambda}; \mathbf{u}) = \mathbb{E}_{\mathcal{D}_0}\big[\ell_0\big(\phi(\mathbf{x}), y\big)\big] + \sum_{i=1}^m \lambda_i \Big[\mathbb{E}_{\mathcal{D}_i}\big[\ell_i\big(\phi(\mathbf{x}), y\big)\big] - u_i\Big]. \tag{3}$$

Based on (3), define dual functions $g(\boldsymbol{\lambda}; \mathbf{u})$ and dual problems $\tilde{\mathsf{D}}^\star(\mathbf{u})$ for given constraint level $\mathbf{u}$,

$$\tilde{\mathsf{D}}^\star(\mathbf{u}) = \max_{\boldsymbol{\lambda} \succeq \mathbf{0}} g(\boldsymbol{\lambda}; \mathbf{u}) = \max_{\boldsymbol{\lambda} \succeq \mathbf{0}} \min_{\phi \in \mathcal{F}} \mathcal{L}(\phi, \boldsymbol{\lambda}; \mathbf{u}). \tag{$\tilde{\mathsf{D}}_{\mathbf{u}}$}$$

While $\tilde{\mathsf{D}}^\star \leq \tilde{\mathsf{P}}^\star$ (weak duality) in general, there are cases in which $\tilde{\mathsf{D}}^\star = \tilde{\mathsf{P}}^\star$ (strong duality), e.g., in convex optimization. The constrained ($\tilde{\mathsf{P}}_{\mathbf{u}}$) is then essentially equivalent to ($\tilde{\mathsf{D}}_{\mathbf{u}}$) that can be tackled by solving an unconstrained, penalized problem (minimizing (3) with respect to $\phi$ and $\mathbf{u}$), while adapting the weights $\lambda_i$ of the penalties (maximizing (3) with respect to $\boldsymbol{\lambda}$). This is, in fact, the basis of operation of primal-dual constrained optimization algorithms [39, Chapter 5]. The penalties $\boldsymbol{\lambda}^\star(\mathbf{u})$ that achieve this equivalence, known as *Lagrange multipliers*, are solutions of ($\tilde{\mathsf{D}}_{\mathbf{u}}$) and subgradients of the perturbation function. Strong duality also holds under the milder Assumption 1 and a constraint qualification requirement stated next.

**Assumption 2 .** There exist a finite relaxation $\mathbf{u} \preceq \infty$ and a function $\phi \in \mathcal{F}$ such that all constraints are met with margin $c > 0$, i.e., $\mathbb{E}_{\mathcal{D}_i}\big[\ell_i(\phi(\mathbf{x}), y)\big] \leq u_i - c$, for all $i = 1, \ldots, m$.

Note that since $\mathbf{u}$ can be large, this requirement is mild. We can thus leverage strong duality to provide an alternative definition of the resilient relaxation in (2).

**Proposition 3.** *Let $\boldsymbol{\lambda}^\star(\mathbf{u})$ be a solution of ($\tilde{\mathsf{D}}_{\mathbf{u}}$) for the relaxation $\mathbf{u}$. Under Assumptions 1 and 2, $\tilde{\mathsf{P}}^\star(\mathbf{u}^*) = \tilde{\mathsf{D}}^\star(\mathbf{u}^*)$ and any $\mathbf{u}^\star \in \mathbb{R}_+^m$ such that $\nabla h(\mathbf{u}^\star) = \boldsymbol{\lambda}^\star(\mathbf{u}^\star)$ is a resilient relaxation of ($\tilde{\mathsf{P}}_{\mathbf{u}}$).*

See appendix D.3 for proof. Whereas Def. 1 specifies the resilience relaxation in terms of the marginal performance gains $\partial \tilde{\mathsf{P}}^\star$, Prop. 3 formulates it in terms of the penalties $\boldsymbol{\lambda}$. Indeed, it establishes that the penalty $\boldsymbol{\lambda}^\star(\mathbf{u}^\star)$ that achieves the resilient relaxation of ($\tilde{\mathsf{P}}_{\mathbf{u}}$) is encoded in the relaxation cost as $\nabla h(\mathbf{u}^\star)$. That is not to say that $h$ directly weighs the requirements as in fixed penalty formulations, since the equilibrium in Prop. 3 also depends on the relative difficulty of satisfying those requirements. Though Prop. 3 does not claim that *all* resilient relaxations satisfy the Lagrange multiplier relation, this holds under additional continuity conditions on ($\tilde{\mathsf{P}}_{\mathbf{u}}$) [40].

Prop. 3 already provides a more computationally amenable definition of resilient relaxation, given that Lagrange multipliers are often computed while solving ($\tilde{\mathsf{P}}_{\mathbf{u}}$), e.g., when using primal-dual methods. Yet, an even more straightforward formulation of Def. 1 is obtained by rearranging (2) as $0 \in \partial(\tilde{\mathsf{P}}^\star + h)(\mathbf{u}^\star)$, as we did in the proof of Prop. 1, which suggests that $\mathbf{u}^\star$ is related to the minimizers of $\tilde{\mathsf{P}}^\star + h$. This is formalized in the following proposition (see proof in appendix D.4).

**Proposition 4.** *A relaxation $\mathbf{u}^\star$ satisfies the resilient equilibrium* (2) *if and only if it is a solution of*

$$\tilde{\mathsf{P}}_\mathsf{R}^\star = \min_{\phi \in \mathcal{F},\, \mathbf{u} \in \mathbb{R}_+^m} \quad \mathbb{E}_{(\mathbf{x},y) \sim \mathcal{D}_0}\Big[ \ell_0\big(\phi(\mathbf{x}), y\big) \Big] + h(\mathbf{u})$$

$$\text{subject to} \quad \mathbb{E}_{(\mathbf{x},y) \sim \mathcal{D}_i}\Big[ \ell_i\big(\phi(\mathbf{x}), y\big) \Big] \le u_i, \quad i = 1, \ldots, m. \tag{$\tilde{\mathsf{P}}$-RES}$$

*The corresponding minimizer $\phi^\star$ is a resilient solution of the functional learning problem $(\tilde{\mathsf{P}}_\mathbf{u})$.*

Prop. 4 shows that it is possible to simultaneously find a resilient relaxation $\mathbf{u}^\star$ and solve the corresponding resilient learning problem. Indeed, a resilient solution of a constrained learning problem can be obtained by incorporating the relaxation cost in its objective. This is reminiscent of first-phase solvers found in interior-point methods used to tackle convex optimization problems [39, Chap. 11]. Note, once again, that this is not the same as directly adding the constraints in the objective as penalties or regularizations. Indeed, recall from Def. 1 that the resilient relaxation balances the marginal effects of $h$ on $\tilde{\mathsf{P}}^\star$ and not $\ell_0$. Before using Prop. 4 to introduce a practical resilient learning algorithm and its approximation and generalization properties (Sec. 4), we use these equivalent formulations to relate resilient learning to classical learning tasks.

### 3.3 Relation to Classical Learning Tasks

**(Un)constrained learning**: Both traditional unconstrained and constrained learning can be seen as limiting cases of resilient learning. Indeed, if $h \equiv 0$, $\mathbf{u}$ has no effect on the objective of $(\tilde{\mathsf{P}}$-RES). We can then take $u_i = B$ for $i = 1, \ldots, m$, which reduces $(\tilde{\mathsf{P}}$-RES) to an unconstrained learning problem (recall that all losses are $[-B, B]$-valued). On the other hand, if $h$ is the indicator function of the non-negative orthant (i.e., $h(\mathbf{u}) = 0$ for $\mathbf{u} \preceq \mathbf{0}$ and $h(\mathbf{u}) = \infty$, otherwise), then it must be that $\mathbf{u} = \mathbf{0}$ in $(\tilde{\mathsf{P}}$-RES) as long as this specification is feasible. Neither of these relaxation costs satisfy the conditions from Def. 1, since they are not componentwise increasing or differentiable, respectively. Still, there exists valid relaxation costs that approximate these problems arbitrarily well (see Appendix D.5).

**Penalty-based methods**: Rather than the constrained formulations in (P) or $(\tilde{\mathsf{P}}_\mathbf{u})$, requirements are often incorporated directly into the objective of learning tasks using fixed penalties as in

$$\underset{\phi \in \mathcal{F}}{\text{minimize}} \ \mathbb{E}_{\mathcal{D}_0}\big[ \ell_0\big(\phi(\mathbf{x}), y\big) \big] + \sum_{i=1}^m \gamma_i \mathbb{E}_{\mathcal{D}_i}\big[ \ell_i\big(\phi(\mathbf{x}), y\big) \big], \tag{4}$$

where the fixed $\gamma_i > 0$ represent the relative importance of the requirements. It is immediate from Prop. 3 that resilient learning with a linear relaxation cost $h(\mathbf{u}) = \sum_i \gamma_i u_i$ is equivalent to (4) as long as $\mathbb{E}_{\mathcal{D}_i}\big[ \ell_i\big(\phi^\star(\mathbf{x}), y\big) \big] \ge 0$. From Def. 1, this is the same as fixing the marginal effect of relaxations on the perturbation function.

**Soft-margin SVM**: Linear relaxation costs are also found in soft-margin SVM formulations, namely

$$\underset{\theta \in \Theta,\, \mathbf{u} \in \mathbb{R}_+^m}{\text{minimize}} \quad \frac{1}{2} \|\theta\|^2 + \gamma \sum_{i=1}^m u_i$$

$$\text{subject to} \quad 1 - y_i \theta^T \mathbf{x}_i \le u_i, \quad i = 1, \ldots, m. \tag{PI}$$

Though written here in its parametrized form, the hypothesis class underlying (PI) (namely, linear classifiers) is convex. It can therefore be seen as an instance of $(\tilde{\mathsf{P}}$-RES) where each loss $\ell_i$ represent a classification requirement on an individual sample. Soft-margin SVM is therefore a resilient learning problem as opposed to its hard-margin version, where $u_i = 0$ for $i = 1, \ldots, m$.

## 4 Resilient Constrained Learning Algorithm

We defined the resilient equilibrium $\mathbf{u}^\star$ in (2) and the equivalent resilient learning problems $(\tilde{\mathsf{P}}_{\mathbf{u}^\star})$ and $(\tilde{\mathsf{P}}$-RES) in the context of the convex functional space $\mathcal{F}$. Contrary to $(\mathsf{P}_\mathbf{u})$, that are defined on the finite dimensional $\mathcal{F}_\theta$, these problems are not amenable to numerical solutions. Nevertheless, we have argued that as long as $\mathcal{F}_\theta$ is a good approximation of $\mathcal{F}$, the values of $(\mathsf{P}_\mathbf{u})$ and $(\tilde{\mathsf{P}}_\mathbf{u})$ are close. We use this idea to obtain a practical primal-dual algorithm (Alg. 1) to approximate solutions

**Algorithm 1** Resilient Constrained Learning ($\eta, \eta_u, \eta_\lambda > 0$; $\theta^{(0)} \in \Theta$; $\boldsymbol{\lambda}^{(0)}, \mathbf{u}^{(0)} \in \mathbb{R}_+^m$)

---

**for** $t = 1, \ldots, T$:
$$\begin{cases} \theta_1 = \theta^{(t-1)} \\ \theta_{n+1} = \theta_n - \eta \nabla_\theta \Big[ \ell_0\big(f_{\theta_n}(\mathbf{x}_{n,0}), y_{n,0}\big) + \sum_{i=1}^m \lambda_i \ell_i\big(f_{\theta_n}(\mathbf{x}_{n,i}), y_{n,i}\big) \Big], \quad n = 1, \ldots, N \\ \theta^{(t)} = \theta_{N+1} \end{cases}$$
$$\mathbf{u}^{(t)} = \Big[ \mathbf{u}^{(t-1)} - \eta_\mathbf{u} \Big( \nabla h\big(\mathbf{u}^{(t-1)}\big) - \boldsymbol{\lambda}^{(t-1)} \Big) \Big]_+$$
$$\lambda_i^{(t)} = \Big[ \lambda_i^{(t-1)} + \eta_\lambda \Big( \tfrac{1}{N} \sum_{n=1}^N \ell_i\big(f_{\theta^{(t-1)}}(\mathbf{x}_{n,i}), y_{n,i}\big) - u_i^{(t-1)} \Big) \Big]_+, \quad i = 1, \ldots, m$$
**end**

---

of (P̃-RES) (and thus (P̃$_{\mathbf{u}^\star}$)). The main result of this section (Thm. 1) establishes how good this approximation can be when using only samples from the $\mathcal{D}_i$.

Explicitly, consider a set of $N$ i.i.d. sample pairs $(\mathbf{x}_{n,i}, y_{n,i})$ drawn from $\mathcal{D}_i$ and define the parametrized, empirical Lagrangian of the resilient learning problem (P̃-RES) as

$$\hat{L}_\theta(\theta, \boldsymbol{\lambda}; \mathbf{u}) = h(\mathbf{u}) + \frac{1}{N} \sum_{n=1}^N \ell_0\big(f_\theta(\mathbf{x}_{n,0}), y_{n,0}\big) + \sum_{i=1}^m \lambda_i \left( \frac{1}{N} \sum_{n=1}^N \ell_i\big(f_\theta(\mathbf{x}_{n,i}), y_{n,i}\big) - u_i \right). \quad (5)$$

The parametrized, empirical dual problem of (P̃-RES) is then given by

$$\hat{D}_R^\star = \max_{\boldsymbol{\lambda} \in \mathbb{R}_+^m} \min_{\theta \in \Theta, \mathbf{u} \in \mathbb{R}_+^m} \hat{L}_\theta(\theta, \boldsymbol{\lambda}; \mathbf{u}). \qquad (\hat{D}\text{-RES})$$

The gap between $\hat{D}_R^\star$ and the optimal value $\tilde{P}_R^\star$ of the original problem ($P_\mathbf{u}$) can be bounded under the following assumptions.

**Assumption 3 .** The loss functions $\ell_i$, $i = 0 \ldots m$, are $M$-Lipschitz continuous.

**Assumption 4 .** For every $\phi \in \mathcal{F}$, there exists $\theta^\dagger \in \Theta$ such that $\mathbb{E}_{\mathcal{D}_i}\big[|\phi(\mathbf{x}) - f_{\theta^\dagger}(\mathbf{x})|\big] \leq \nu$, for all $i = 0, \ldots, m$.

**Assumption 5 .** There exists $\xi(N, \delta) \geq 0$ such that for all $i = 0, \ldots, m$ and all $\theta \in \Theta$,

$$\left| \mathbb{E}_{\mathcal{D}_i}\big[\ell_i(f_\theta(\mathbf{x}), y)\big] - \frac{1}{N} \sum_{n=1}^N \ell_i\big(f_\theta(\mathbf{x}_{n,i}), y_{n,i}\big) \right| \leq \xi(N, \delta)$$

with probability $1 - \delta$ over draws of $\{(\mathbf{x}_{n,i}, y_{n,i})\}$.

Although the parameterized functional space $\mathcal{F}_\theta$ can be non-convex, as is the case for neural networks, Ass. 4 requires that it is rich in the sense that the distance to its convex hull is bounded. When $\xi$ exists for all $\delta > 0$ and is a decreasing function of $N$, Ass. (5) describes the familiar uniform convergence from learning theory. Such generalization bounds can be derived based on bounded VC dimension, Rademacher complexity, or algorithmic stability, to name a few [41].

We can now state the main result of this section, whose proof is provided in Appendix C.

> **Theorem 1.** *Consider $\tilde{P}_R^\star$ and $\mathbf{u}^\star$ from (P̃-RES) and $\hat{D}_R^\star$ from (D̂-RES). Under Ass. 1–5, it holds with probability of $1 - (3m + 2)\delta$ that*
>
> $$\left| \tilde{P}_R^\star - \hat{D}_R^\star \right| \leq h(\mathbf{u}^\star + \mathbb{1} \cdot M\nu) - h(\mathbf{u}^\star) + M\nu + (1 + \Delta)\xi(N, \delta). \qquad (6)$$

Thm. 1 suggests that resilient learning problems can be tackled using (D̂-RES), which can be solved using saddle point methods, as in Alg. 1. Even if the inner minimization problem is non-convex, dual ascent methods can be shown to converge as long as its solution can be well approximated using stochastic gradient descent [38, Thm. 2], as is often the case for overparametrized NNs [42–44]. A more detailed discussion on the algorithm can be found in Appendix E. Next, we showcase its performance and our theoretical results in two learning tasks.

# 5 Numerical Experiments

We now showcase the numerical properties of resilient constrained learning. As illustrative case studies, we consider federated learning under class imbalance and invariance constrained learning. Additional experimental results for both setups are included in Appendix F and H. We also include ablations on the hyperparameters of the method, including dual and perturbation learning rates, and the choice of the perturbation cost function $h$.

## 5.1 Heterogeneous Federated Learning

Federated learning [45] entails learning a common model in a distributed manner by leveraging data samples from different *clients*. Usually, average performance across all clients is optimized under the assumption that data from different clients is identically distributed. In practice, heterogeneity in local data distributions can lead to uneven performance across clients [46, 47]. Since this may be undesirable, a sensible requirement in this setting is that the loss of the model is *similar* for all clients.

Let $\mathfrak{D}_c$ be the distribution of data pairs for Client $c$, and $R_c(f_{\boldsymbol{\theta}}) = \mathbb{E}_{(\mathbf{x},y)\sim\mathfrak{D}_c}[\ell(f_{\boldsymbol{\theta}}(\mathbf{x}),y)]$ its statistical risk. We denote the average performance as $\overline{R}(f_{\boldsymbol{\theta}}) := (1/C)\sum_{i=1}^{C} R_c(f_{\boldsymbol{\theta}})$, where $C$ is the number of clients. As proposed in [18] heterogeneity issues can be tackled by imposing a proximity constraint between the performance of each client $R_c$, and the loss averaged over all clients $\overline{R}$. This leads to the constrained learning problem

$$\min_{\boldsymbol{\theta}\in\Theta}. \quad \overline{R}(f_{\boldsymbol{\theta}}) \tag{P-FL}$$
$$\text{s. to} \quad R_c(f_{\boldsymbol{\theta}}) - \overline{R}(f_{\boldsymbol{\theta}}) - \epsilon \leq 0, \qquad c = 1,\ldots,C,$$

where $\epsilon > 0$ is a small (fixed) positive scalar. It is ready to see that this problem is of the form in (P); see Appendix F. Note that the constraint in (P-FL) is an asymmetric (and relaxed) version of the equality of error rates common in fairness literature.

As shown by [18] this problem can be solved via a primal-dual approach in a privacy-preserving manner and with a negligible communication overhead, that involves sharing dual variables. However, because the heterogeneity of the data distribution across clients is unknown *a priori*, it can be challenging to specify a single constraint level $\epsilon$ for all clients that results in a reasonable trade-off between overall and individual client performance. In addition, the performance of all clients may be significantly reduced by a few clients whose poor performance make the constraint hard to satisfy. The heuristic adopted in [18] is to clip dual variables that exceed a fixed value.

In contrast, we propose to solve a resilient version of (P-FL) by including a relaxation $u_c, c = 1,\ldots,C$ for each constraint and a quadratic relaxation cost $h(\mathbf{u}) = \alpha\|\mathbf{u}\|_2^2$. The resilient version of problem (P-FL) can also be solved in a privacy preserving manner as long as $h(\mathbf{u})$ is separable in each of the constraint perturbations $u_c$. In that case $u_c$ can be updated locally by client $c$.

Following the setup of [18], heterogeneity across clients is generated through class imbalance. More specifically, samples from different classes are distributed among clients using a Dirichlet distribution. Experimental and algorithmic details along with privacy considerations are presented in Appendix F.

**Constraint relaxation and relative difficulty:** Our approach relaxes constraints according to their relative difficulty. In the context of class imbalance, the majority classes exert a stronger influence on the overall objective (average performance). Therefore, the overall performance $\overline{R}$ tends to favor smaller losses in the majority classes rather than in the minority ones. As a result, meeting the proximity constraint in (P-FL) for clients whose datasets contain a higher fraction of training samples from the minority classes can deteriorate performance. In Figure 2(left), we demonstrate that the constraint is effectively relaxed more for these clients.

**Controlling the performance vs. relaxation trade-off:** Through the choice of the relaxation cost function, we can control the trade-off between relaxing the equalization requirements from (P-FL) and performance. To illustrate this, we perform an ablation on the coefficient $\alpha$ in the quadratic relaxation cost $h(\mathbf{u}) = \alpha\|\mathbf{u}\|_2^2$. As sown in Figure 2(middle) smaller values of $\alpha$ enable better performance at the cost of larger relaxations. As $\alpha$ increases, the relaxations vanish and the problem approaches the original constrained problem. In this manner, the resilient approach enables navigating this trade-off

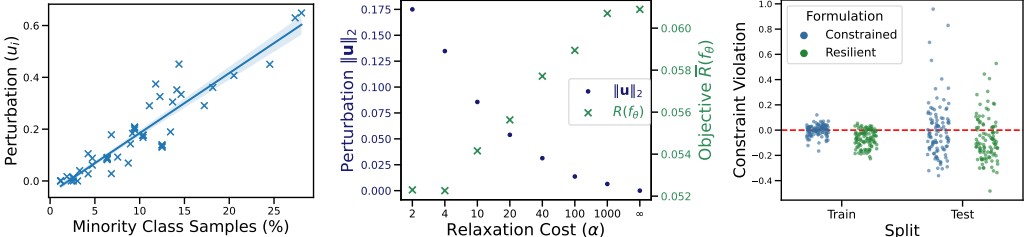

Figure 2: (Left) Constraint relaxation and relative difficulty for federated learning under heterogeneous class imbalance across clients (crosses). We plot the perturbation $u_c$ against the fraction of the dataset of client $c$ from the minority class, which is associated to how difficult it is to satisfy the constraint, since minority classes typically have higher loss. (Middle) Relaxation cost parameter ($h(\mathbf{u}) = \alpha\|\mathbf{u}\|_2^2$) vs. final training loss and perturbation norm. (Right) Constraint violations on train and test sets

by changing a single hyperparameter. Still, the optimal relaxation for each client is determined by their local data distribution, i.e., the relative difficulty of the constraint. In Appendix F.4 we also perform an ablation on the cost function by changing the perturbation norm.

**Constraint violation and generalization:** Relaxing stringent requirements not only makes the empirical problem easier to solve, but it can also lead to a better empirical approximation of the underlying statistical problem. As shown in Figure 2(right), the resilient approach has a smaller fraction of clients that are infeasible at the end of training. In addition, when constraints are evaluated on the test set, larger constraint violations are observed for some (outlier) clients in the hard constrained approach ($u_c = 0$). In Appendix F.4 we show that this holds across different problem settings. In addition, we also illustrate the fact that this is partly due to generalization issues, i. e., that overly stringent requirements can harm generalization. This observation is in line with the constrained learning theory developed in [7, 8].

**Comparing client performances:** Our method should sacrifice the performance of outliers–which represent hard to satisfy constraints–in order to benefit the performance of the majority of the clients. In order to showcase this, we sort clients according to their test accuracy $\text{Acc}_{[1]} \geq \text{Acc}_{[2]} \geq \ldots \geq \text{Acc}_{[c]}$. We report the fraction of clients in the resilient method that outperform equally ranked clients for the constrained baseline [18] ($\text{Acc}_{[c]}^{\text{res}} \geq \text{Acc}_{[c]}^{\text{const}}$), as well as the average and maximum increases and decreases in performance. As shown in Table 1, the majority of clients achieve a small increase in performance, while a small fraction experiences a larger decrease. In Appendix F.4 we include plots of ordered accuracies to further illustrate this.

| Dataset | Imbalance Ratio | Improved over constrained (%) | Max Improvement | Max Deterioration |
|---------|----------------|-------------------------------|-----------------|-------------------|
| CIFAR10 | 10 | 77 | 1.5 | 10.0 |
|         | 20 | 79 | 2.1 | 9.1 |
| F-MNIST | 10 | 92 | 4.8 | 23.3 |
|         | 20 | 94 | 2.6 | 19.4 |

Table 1: Changes in accuracy for equally ranked clients for the resilient method compared to the constrained baseline [18]. We report the fraction of equally ranked clients which improved their accuracy, along with the mean and maximum change across all clients that improve (imp.) and decrease (dec.) their accuracy, respectively. Performance improves for most clients, although at the cost of a decrease in performance for a few outliers

## 5.2 Invariance Constrained Learning

As in [21], we impose a constraint on the loss on transformed inputs, as a mean of achieving robustness to transformations which may describe invariances or symmetries of the data. Explicitly,

$$\min_{\boldsymbol{\theta}\in\Theta}. \quad \mathbb{E}_{(\mathbf{x},y)\sim\mathfrak{D}}\left[\ell(f_{\boldsymbol{\theta}}(\mathbf{x}),y)\right] \tag{P-IC}$$

$$\text{s. to} \quad \mathbb{E}_{(\mathbf{x},y)\sim\mathfrak{D}}\left[\max_{g\in\mathcal{G}_i}\ell(f_{\boldsymbol{\theta}}(g\mathbf{x}),y)\right] - \epsilon \leq 0, \qquad i = 1,\ldots,m,$$

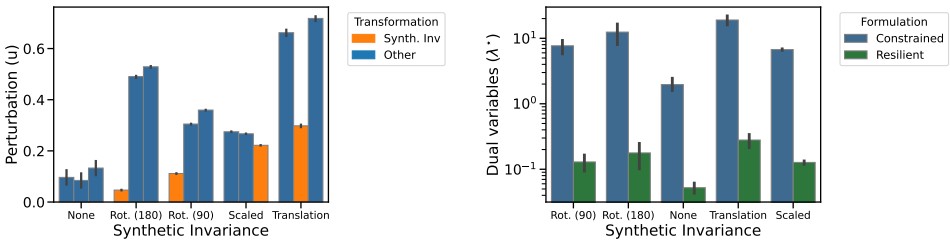

Figure 3: (Left) Constraint Relaxation and Relative difficulty for Synthetically invariant datasets. Bars denote the perturbation $\mathbf{u}_i$ associated with different transformations, and the x-axis shows the synthetic invariance of the dataset. (Right) Sensitivity (Dual variables) for Synthetically invariant datasets. Bars correspond to the values of dual variables (for all constraints) at the end of training. We compare the resilient and constrained approach accross different synthetic datasets.

where $\mathcal{G}_i$ is a set of transformations $g : \mathcal{X} \to \mathcal{X}$ such as image rotations, scalings, or translations. The constraint can be approximated by sampling augmentations using MCMC methods (see [12, 21] for details). However, it is challenging to specify the transformation sets $\mathcal{G}_i$ and constraint levels $\epsilon_i$ adequately, since they depend on the invariances of the unknown data distribution and whether the function class is sufficiently expressive to capture these invariances. Therefore, we propose to solve a resilient version of problem (P-IC) using $h(\mathbf{u}) = \alpha \|\mathbf{u}\|_2^2$ as a perturbation cost.

We showcase our approach on datasets with artificial invariances, following the setup of [23]. Explicitly, we generate the synthetic datasets, by applying either rotations, translations, or scalings, to each sample in the MNIST [48] and FashionMNIST [49] datasets. The transformations are sampled from uniform distributions over the ranges detailed in Appendix H.

**Constraint relaxation and relative difficulty** As shown in Figure 3(left) the relaxations ($\mathbf{u}$) associated with synthetic invariances are considerably smaller than for other transformations. This indicates that when the transformations in the constraint correspond to a true invariance of the dataset, the constraint is easier to satisfy. On the other hand, when the transformations are not associated with synthetic invariances of the dataset, they are harder to satisfy and thus result in larger relaxations. This results in smaller dual variables, as shown in Figure 3 (right).

**Resilience Improves Performance** The resilient approach is able to handle the misspecification of invariance requirements, outperforming in terms of test accuracy both the constrained and unconstrained approaches. We include results for MNIST in Table 2 and F-MNIST in Appendix H. In addition, though it was not designed for that purpose, our approach shows similar performance to the invariance learning method Augerino [22].

| Dataset | Method | Rotated (180) | Rotated (90) | Translated | Scaled | Original |
|---------|--------|---------------|--------------|------------|--------|----------|
| MNIST | Augerino | **97.78 ± 0.03** | 96.38 ± 0.00 | 94.65 ± 0.01 | 97.53 ± 0.00 | 98.44 ± 0.00 |
| | Unconstrained | 94.49 ± 0.12 | 96.25 ± 0.13 | 94.64 ± 0.20 | 97.47 ± 0.03 | 98.45 ± 0.06 |
| | Constrained | 94.55 ± 0.18 | 96.90 ± 0.07 | 93.74 ± 0.07 | 97.92 ± 0.15 | 98.74 ± 0.08 |
| | Resilient | 95.38 ± 0.18 | **97.19 ± 0.09** | **95.21 ± 0.15** | **98.20 ± 0.04** | **98.86 ± 0.02** |

Table 2: Classification accuracy for synthetically invariant MNIST. We use the same invariance constraint level $\epsilon_i = 0.1$ for all transformations. We include the invariance learning method Augerino [22] as a baseline.

## 6 Conclusion

This paper introduced a method to specify learning constraints by balancing the marginal decrease in the objective value obtained from relaxation with the marginal increase in a relaxation cost. This resilient equilibrium has the effect of prioritizing the relaxation of constraints that are harder to satisfy. The paper also determined conditions under which this equilibrium exists and provided an algorithm to automatically find it during training, for which approximation and statistical guarantees were derived. Experimental validations showcased the advantages of resilient constrained learning for classification with invariance requirements and federated learning. Future work includes exploring different relaxation costs and applications to robustness against disturbances and outliers.

## Acknowledgments and Disclosure of Funding

The work of A. Ribeiro and I. Hounie is supported by NSF-Simons MoDL, Award 2031985, NSF AI Institutes program, Award 2112665, and NSF HDR TRipods Award 1934960. The work of Dr. Chamon is supported by the Deutsche Forschungsgemeinschaft (DFG, German Research Foundation) under Germany's Excellence Strategy (EXC 2075-390740016).

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

# A    Constrained learnability and relaxations

In this section, we illustrate how constraints affect learnability, an issue that can be exacerbated by relaxing their specification either too much or too little. Resilient constrained learning, on the other hand, progressively adapts to the underlying problem, striking a balance between modifying the learning task and improving the objective value that can overcome these issues.

Let us begin by stating what we mean by constrained learnability (see [38, Def. 4] for details).

**Definition 2 (PACC).** A hypothesis class $\mathcal{F}_\theta$ is *probably approximately correct constrained* (PACC) learnable with respect to the losses $\{\ell_i\}_{i=0,\ldots,m}$ if there exists an algorithm that, for every $\epsilon, \delta \in (0,1)$ and every distribution $\mathcal{D}_i$, $i = 0, \ldots, m$, can obtain $f_\theta \in \mathcal{F}_\theta$ using $N(\epsilon, \delta, m)$ samples from each $\mathcal{D}_i$ that is, with probability $1 - \delta$,

1) probably approximately optimal, i.e., for $\mathsf{P}^\star$ as in (P),

$$\left| \mathbb{E}_{(\mathbf{x},y) \sim \mathcal{D}_0} \left[ \ell_0 \big( f_\theta(\mathbf{x}), y \big) \right] - \mathsf{P}^\star \right| \le \epsilon, \text{ and} \tag{7}$$

2) probably approximately feasible, i.e.,

$$\mathbb{E}_{(\mathbf{x},y) \sim \mathcal{D}_i} \left[ \ell_i \big( f_\theta(\mathbf{x}), y \big) \right] \le \epsilon, \quad \text{for all } i \ge 1. \tag{8}$$

Definition 2 is an extension of the probably approximately correct (PAC) framework from classical learning theory to the problem of learning under requirements. Indeed, note that for unconstrained learning problems [$m = 0$ in (P)], $\mathsf{P}^\star \le \mathbb{E}_{\mathcal{D}_0} \left[ \ell_0 \big( f_\theta(\mathbf{x}), y \big) \right]$ for all $\theta \in \Theta$. In that case, (7) reduces to the classical definition of PAC learnability [41, Def. 2.14]. In general, however, a PACC learner must also satisfy the approximate feasibility condition (8).

Constrained learning theory shows that, under mild conditions, the PAC learnability of $\mathcal{F}_\theta$ with respect to each individual loss $\ell_i$ implies its PACC learnability [38, Thm. 1]. This does not mean that an empirical version of (P) is a PACC learner, i.e., approximates the value of $\mathsf{P}^\star$. In fact, this is typically not the the case.

Case in point, consider the learning problem

$$\mathsf{P}_e^\star = \min_{\theta \in \Theta} \quad J(\theta) \triangleq \mathbb{E}_{\mathcal{D}_0} \big[ |\theta^\top \mathbf{x}| \big]$$
$$\text{subject to} \quad \mathbb{E}_{\mathcal{D}_1} \big[ y \theta^\top \mathbf{x} \big] \le -1, \quad \mathbb{E}_{\mathcal{D}_2} \big[ y \theta^\top \mathbf{x} \big] \le 1, \tag{PII}$$

where $\mathcal{D}_0$ is the distribution of

$$(\mathbf{x}, y) = \begin{cases} \big( [\tau, -\tau]^T, -1 \big), & \text{with prob. } 1/2 \\ \big( [0, \alpha]^T, 1 \big), & \text{with prob. } 1/2 \end{cases},$$

$\mathcal{D}_1$ is such that $(\mathbf{x}, y) = ([-1, \tau], 1)$, and $\mathcal{D}_2$ is such that $(\mathbf{x}, y) = ([-\tau, 1], 1)$, where $\alpha$ is drawn uniformly at random from $[0, 1/4]$ and $\tau$ is drawn uniformly at random from $[-1/2, 1/2]$. Notice that the $\mathcal{D}_i$ are correlated through the random variable $\tau$. The parameters $\theta = [\theta_1, \theta_2]$ are taken from the finite set $\Theta = \{\theta^a, \theta^b, \theta^c, \theta^d\}$, described below together with their corresponding objective values:

$$J(\theta) = \begin{cases} 1/32, & \theta = \theta^a \triangleq [1/2, 1/2]^T \\ 1/16, & \theta = \theta^b \triangleq [1, 1]^T \\ 1/16 + 1/24, & \theta = \theta^c \triangleq [1, 1/3]^T \\ 1/8, & \theta = \theta^d \triangleq [1, 0]^T \end{cases}. \tag{9}$$

Notice that under these distributions, the constraints in (PII) reduce to $-\theta_1 \le -1$ and $\theta_2 \le 1$. Hence, all parameters expect for $[1/2, 1/2]^T$ are feasible for the statistical (PII). From (9), its optimal value is therefore $\mathsf{P}_e^\star = 1/16$ achieved for $\theta^\star = \theta^b$.

In the sequel, we will show that (i) the empirical version of (PII) [as in (PIII)] recovers its solution with exponentially small probability (Sec. A.1); (ii) relaxing the empirical problem [as in (PIV)] too little or too much fails to mitigate this issue (Sec. A.2); and (iii) a resilient relaxation [as in (PV)] allows the empirical version of (PII) to recover its solutions with high probability. In summary, the empirical counterpart of (PII) can be relaxed into a PACC learner, but care must be taken when choosing that relaxations.

## A.1 Empirical constrained risk minimization (ECRM)

Consider the empirical version of (9), which can be written as

$$\hat{\mathsf{P}}_e^\star = \min_{\theta \in \Theta} \quad \hat{J}(\theta) \triangleq \frac{1}{N} \sum_{n=1}^N |\theta^\top \mathbf{x}_n| \tag{PIII}$$
$$\text{subject to} \quad \theta_1 \geq 1 + \bar{\tau}\theta_2, \quad \theta_2 \leq 1 + \bar{\tau}\theta_1,$$

where $\bar{\tau} = \frac{1}{N} \sum_{n=1}^N \tau_n$ is the empirical average of i.i.d. samples $\tau_n$ drawn uniformly at random from $[-1/2, 1/2]$. Notice that

- for $\theta^a$, the first constraint reduces to $1/2 \geq 1 + \bar{\tau}/2$ and since $\bar{\tau} \geq -1/2$, it is violated unless $\bar{\tau} = 0$; and
- for $\theta^b$, the constraints read $1 \geq 1 + \bar{\tau}$ and $1 \leq 1 + \bar{\tau}$, so that at least one is violated unless $\bar{\tau} = 0$.

Since $\tau$ is a continuous distribution, $\bar{\tau} \neq 0$ almost surely. Immediately, we obtain that $\hat{\mathsf{P}}_e^\star \geq \min(\hat{J}(\theta^c), \hat{J}(\theta^d))$. Additionally, $\hat{\mathsf{P}}_e^\star < \infty$ since $\theta^d$ is always feasible. Hence,

$$\mathbb{P}\big[|\hat{\mathsf{P}}_e^\star - \mathsf{P}_e^\star| \leq 1/64\big] \leq \mathbb{P}\big[|\hat{J}(\theta^c) - \mathsf{P}_e^\star| \leq 1/64 \cup |\hat{J}(\theta^d) - \mathsf{P}_e^\star| \leq 1/64\big]$$
$$\leq \mathbb{P}\big[|\hat{J}(\theta^c) - \mathsf{P}_e^\star| \leq 1/64\big] + \mathbb{P}\big[|\hat{J}(\theta^d) - \mathsf{P}_e^\star| \leq 1/64\big]$$
$$\leq \mathbb{P}\big[|\hat{J}(\theta^c) - J(\theta^c)| \geq 1/24 - 1/64\big] + \mathbb{P}\big[|\hat{J}(\theta^d) - J(\theta^d)| \geq 1/16 - 1/64\big]$$
$$\leq 4e^{-0.001N}.$$

## A.2 Relaxed ECRM

Consider now a relaxation of (PIII), namely

$$\hat{\mathsf{P}}_x^\star = \min_{\theta \in \Theta} \quad \hat{J}(\theta) \triangleq \frac{1}{N} \sum_{n=1}^N |\theta^\top \mathbf{x}_n| \tag{PIV}$$
$$\text{subject to} \quad \theta_1 \geq 1 + \bar{\tau}\theta_2 - u_1, \quad \theta_2 \leq 1 + \bar{\tau}\theta_1 + u_2.$$

Problem (PIV) is the empirical version of $(\mathsf{P_u})$ for the relaxation $\mathbf{u} = [u_1, u_2]^T$. It is straightforward to see that if $u_1, u_2 < \bar{\tau}$, then we are in the situation of Sec. A.1 and once again $\mathbb{P}\big[|\hat{\mathsf{P}}_e^\star - \mathsf{P}_e^\star| \leq 1/64\big] \to 0$ as $N \to \infty$.

On the other hand, notice that the constraints always hold whenever $u_1 \geq 1 + \bar{\tau}\theta_2 - \theta_1$ and $u_2 \geq \theta_2 - \bar{\tau}\theta_1 - 1$:

- For $\theta^a$, this reduces to $u_1 \geq (\bar{\tau} + 1)/2$, since $u_2 \geq -(\bar{\tau} + 1)/2$ is satisfied for all $\mathbf{u} \succeq \mathbf{0}$. We therefore obtain that

$$0 \leq \hat{\mathsf{P}}_x^\star \leq \hat{J}(\theta^a), \quad \text{for } u_1 \geq \frac{\bar{\tau} + 1}{2}.$$

In this case, recalling that $\bar{\tau} \neq 0$ almost surely yields

$$\mathbb{P}\big[|\hat{\mathsf{P}}_x^\star - \mathsf{P}_e^\star| \leq 1/64\big] = \mathbb{P}\big[|\hat{J}(\theta^a) - 1/16| \leq 1/64 \cap \hat{J}(\theta^a) \leq \hat{J}(\theta^b) \vee \hat{J}(\theta^c) \vee \hat{J}(\theta^d))\big]$$
$$\leq \mathbb{P}\big[|\hat{J}(\theta^a) - 1/16| \leq 1/64\big],$$

where we use $x \vee y = \min(x, y)$. Using the fact that $J(\theta^a) = 1/32$, we obtain

$$\mathbb{P}\big[|\hat{\mathsf{P}}_x^\star - \mathsf{P}_e^\star| \leq 1/64\big] \leq \mathbb{P}\big[|\hat{J}(\theta^a) - J(\theta^a) - 1/32| \leq 1/64\big]$$
$$\leq \mathbb{P}\big[|\hat{J}(\theta^a) - J(\theta^a)| \geq 1/64\big] \leq 2e^{-0.0001N}.$$

- For $\theta^b$, this becomes $u_1 \geq \bar{\tau}$ and $u_2 \geq \bar{\tau}$. Hence,

$$\hat{J}(\theta^a) \neq \hat{\mathsf{P}}_x^\star \leq \hat{J}(\theta^b), \quad \text{for } \bar{\tau} \leq u_1 < \frac{\bar{\tau} + 1}{2} \text{ and } u_2 \geq \bar{\tau}.$$

Since $P_e^\star = J(\theta^b)$, we therefore get

$$\mathbb{P}\big[|\hat{P}_x^\star - P_e^\star| \leq 1/64\big] = \mathbb{P}\big[|\hat{J}(\theta^b) - J(\theta^b)| \leq 1/64 \cap \hat{J}(\theta^b) \leq \hat{J}(\theta^c) \vee \hat{J}(\theta^d)\big]$$
$$\geq \mathbb{P}\big[\hat{J}(\theta^b) \leq \hat{J}(\theta^c) \vee \hat{J}(\theta^d)\big] - \mathbb{P}\big[|\hat{J}(\theta^b) - J(\theta^b)| \geq 1/64\big]$$

Notice that since $\hat{J}(\theta)$ concentrates around $J(\theta)$ for $\theta \in \Theta$, for every $\epsilon > 0$, there exists $N_0$ such that $\mathbb{P}\big[\hat{J}(\theta^b) \leq \hat{J}(\theta^c) \vee \hat{J}(\theta^d)\big] \geq 1 - \epsilon$. We therefore conclude that

$$\mathbb{P}\big[|\hat{P}_x^\star - P_e^\star| \leq 1/64\big] \geq 1 - \epsilon - \mathbb{P}\big[|\hat{J}(\theta^b) - J(\theta^b)| \geq 1/64\big]$$
$$\geq 1 - \epsilon - 2e^{-0.0004N}$$

In summary, we have that

$$\mathbb{P}\big[|\hat{P}_x^\star - P_e^\star| \leq 1/64\big] = \begin{cases} \leq 4e^{-0.001N}, & u_1, u_2 < \bar{\tau} \\ \geq 1 - 3e^{-Nt^2}, & \bar{\tau} \leq u_1 < \dfrac{\bar{\tau}+1}{2} \text{ and } u_2 \geq \bar{\tau} \\ \leq 2e^{-0.0001N}, & u_1 \geq \dfrac{\bar{\tau}+1}{2} \end{cases}$$

## A.3   Resilient ECRM

Finally, consider the resilient version of (PIII)

$$\hat{P}_R^\star = \min_{\theta \in \Theta,\, \mathbf{u} \in \mathbb{R}_+^m} \quad \hat{J}(\theta) + \frac{\|\mathbf{u}\|^2}{2} \tag{PV}$$
$$\text{subject to} \quad \theta_1 \geq 1 + \bar{\tau}\theta_2 - u_1, \quad \theta_2 \leq 1 + \bar{\tau}\theta_1 + u_2$$

and let $\hat{\theta}_R^\star$ be a solution of (PV). Notice that (PV) is an empirical version of the equivalent formulation in Prop. 4. We consider the value of the problem for each possible element of $\Theta$:

- For $\theta^a$ to be feasible, it must be that $u_1 \geq (\bar{\tau}+1)/2$ ($u_2$, on the other hand, can be 0). Hence, we have that
$$\hat{P}_R^\star \leq \hat{J}(\theta^a) + \frac{(\bar{\tau}+1)^2}{8}.$$

- For $\theta^b$ to be feasible, it must be that $u_1 \geq \bar{\tau}$ and $u_2 \geq \bar{\tau}$. Hence,
$$\hat{P}_R^\star \leq \hat{J}(\theta^b) + \bar{\tau}^2.$$

- For $\theta^c$ to be feasible, it must be that $u_1 \geq \bar{\tau}/3$ (once again the second constraint need not be relaxed). Hence,
$$\hat{P}_R^\star \leq \hat{J}(\theta^c) + \frac{2\bar{\tau}^2}{9}.$$

- Finally, $\theta^d$ is feasible for all $\mathbf{u} \in \mathbb{R}_+^m$. Hence,
$$\hat{P}_R^\star \leq \hat{J}(\theta^d).$$

Putting together these conditions, we obtain that for $\theta^b$ to be optimal, i.e., for $\hat{J}(\hat{\theta}_R^\star) \to P_e^\star$, it must be that

$$\hat{J}(\theta^b) + \bar{\tau}^2 \leq \hat{J}(\theta^a) + \frac{(\bar{\tau}+1)^2}{8} \vee \hat{J}(\theta^c) + \frac{2\bar{\tau}^2}{9} \vee \hat{J}(\theta^d) \tag{10}$$

From the fact that $\hat{J}(\theta)$ concentrates around $J(\theta)$ for $\theta \in \Theta$ (e.g., by Hoeffding's inequality) and using the values of $J$ from (9), it holds that there exists $N_0$ such that $\theta^b$ is optimal for $N \geq N_0$ as long as $|\bar{\tau}| \leq 0.2$. We therefore conclude that

$$\mathbb{P}\big[|J(\hat{\theta}_R^\star) - P_e^\star| \leq 1/64\big] \geq \mathbb{P}\big[\hat{\theta}_R^\star = \theta^b\big] \geq \mathbb{P}\big[|\bar{\tau}| \leq 0.2\big] \geq 1 - 2e^{-0.08N}.$$

# B Convexity of the Perturbation Function for Nonconvex Losses

The definition of resilient equilibrium and all subsequent results in the paper require convexity of the perturbation function $\tilde{P}^\star(\mathbf{u})$ defined in ($\tilde{P}_\mathbf{u}$). This requisite is stated in Assumption 1.

Assumption 1 is well known to hold if the optimization problem ($\tilde{P}_\mathbf{u}$) is convex, for which it is sufficient for the class $\mathcal{F}$ to be a convex set and the losses $\ell_i$ to be convex functions (see, e.g., [40]). If the problem is not parametric, as is, e.g., (P), then assuming the hypothesis class $\mathcal{F}$ is convex is mild. Consider, e.g., the set $\mathcal{F}$ to be the intersection of the space of $\mathcal{D}_i$-measurable functions. Overall, it is not a significant restriction to assume that the losses are convex since this is the case for typical losses, such as cross entropy and mean squared error[1].

This convex losses requirement can be relaxed as long as the distributions are well-behaved and the hypothesis class $\mathcal{F}$ is rich enough. We define these conditions in two technical assumptions. In what follows, we consider the distributions $\mathcal{D}_i$ are defined over the same probability space $(\mathcal{X} \times \mathcal{Y}, \Sigma, \mathbb{P}_i)$ for the sigma-algebra $\Sigma$ and the measures $\mathbb{P}_i$.

**Assumption 6 (Non-atomic distributions).** The set $\mathcal{Y}$ is finite and the conditional random variables $\mathbf{x} \mid y$ induced by the $\mathcal{D}_i$ measures are non-atomic. That is, for every measurable event $Z \in \Sigma$ with strictly positive conditional measure $\mathbb{P}_{i,\mathbf{x}|y}(Z) > 0$, there exist an event $\Sigma \ni Z' \subset Z$ strictly included in $Z$ with positive conditional measure $mbP_{i,\mathbf{x}|y}(Z') > 0$.

**Assumption 7 (Decomposable hypothesis class).** The set $\mathcal{F}$ is decomposable in the sense that for all pair of functions $\phi_1, \phi_2 \in \mathcal{F}$ and all measurable set $Z \in \Sigma$, it holds that $\bar{\phi} \in \mathcal{F}$ for

$$\bar{\phi}(x) = \begin{cases} \phi_1(x), & x \in Z \\ \phi_2(x), & x \notin Z \end{cases}$$

Assumption 6 requires data probability distributions in which all points have zero measure. Assumption 7 requires hypothesis classes that are closed under "cutting and stitching" of functions.

In turns out that Assumptions 6 and 7 are enough to guarantee that the perturbation function is convex. In particular, they imply the following result [38, Lemma A.1].

**Lemma 1.** *Define the cost constraint epigraph of* ($\tilde{P}_\mathbf{u}$) *to be the set*

$$\mathcal{C} = \left\{ [s_0; \mathbf{s}] \in \mathbb{R}^{m+1} \mid \exists \phi \in \mathcal{F} \text{ such that } \mathbb{E}_{\mathcal{D}_i} \left[ \ell_i(\phi(\mathbf{x}), y) \right] \le s_i \ i = 0, \dots, m \right\}. \quad (11)$$

*Under Assumptions 6 and 7, the set $\mathcal{C}$ is convex.*

Notice that Lemma 1 does not require the losses $\ell_i$ to be convex for the cost-constraint epigraph to be convex. This is a significant fact since it implies that the perturbation function of ($\tilde{P}_\mathbf{u}$) is also convex, as we show next.

**Proposition 5.** *Consider the perturbation function $\tilde{P}^\star(\mathbf{u})$ defined in* ($\tilde{P}_\mathbf{u}$). *Under Assumptions 6 and 7, $\tilde{P}^\star(\mathbf{u})$ is convex.*

*Proof.* Let $\mathbf{v}$ and $\mathbf{w}$ be two perturbations of ($\tilde{P}_\mathbf{u}$) and define the convex combination perturbation $\mathbf{u}_a = a\mathbf{v} + (1-a)\mathbf{w}$ for some $a \in [0,1]$. To prove that $\tilde{P}^\star(\mathbf{u})$ is convex we need to show that $\tilde{P}^\star(\mathbf{u}_a) \le a\tilde{P}^\star(\mathbf{v}) + (1-a)\tilde{P}^\star(\mathbf{w})$ for all $a$.

If either $\mathbf{v}$ or $\mathbf{w}$ yield infeasible problems we have $\tilde{P}^\star(\mathbf{v}) = \infty$ or $\tilde{P}^\star(\mathbf{w}) = \infty$. In this case, $\tilde{P}^\star(\mathbf{u}_a) \le a\tilde{P}^\star(\mathbf{v}) + (1-a)\tilde{P}^\star(\mathbf{w}) = \infty$. In the more interesting case where $\mathbf{v}$ and $\mathbf{w}$ both yield feasible problems, we have that

$$[\tilde{P}^\star(\mathbf{v}); \mathbf{v}] \in \mathcal{C} \quad \text{and} \quad [\tilde{P}^\star(\mathbf{w}); \mathbf{w}] \in \mathcal{C}, \quad (12)$$

for $\mathcal{C}$ defined in (11). Indeed, any $\phi_\mathbf{v}^\star$ that solves ($\tilde{P}_\mathbf{v}$) must satisfy $\mathbb{E}_{\mathfrak{D}_i} \left[ \ell_i(\phi_\mathbf{v}^\star(\mathbf{x}), y) \right] \le u_i$ for $i > 0$ and $\mathbb{E}_{\mathfrak{D}_0} \left[ \ell_0(\phi_\mathbf{v}^\star(\mathbf{x}), y) \right] = \tilde{P}^\star(\mathbf{v})$. Likewise for a function $\phi_\mathbf{w}^\star$ that solves ($\tilde{P}_\mathbf{w}$).

Under Assumptions 6 and 7, we can combine (12) and Lemma 1 to get that

$$a[\tilde{P}^\star(\mathbf{v}); \mathbf{v}] + (1-a)[\tilde{P}^\star(\mathbf{w}); \mathbf{w}] \in \mathcal{C}, \quad \text{for all } a \in [0,1]. \quad (13)$$

---

[1]It is only when dealing with the parametrized $\mathcal{F}_\theta$ that the composition of the parametrization with the loss may lead to a convex function of the decision variable $\theta$.

For this to hold, the definition of $\mathcal{C}$ in (11) implies that there exists a function $\phi_a \in \mathcal{F}$ such that

$$\mathbb{E}\left[\ell_0(\phi_a(\mathbf{x}), y)\right] \leq a\tilde{\mathsf{P}}^\star(\mathbf{v}) + (1-a)\tilde{\mathsf{P}}^\star(\mathbf{w}) \quad \text{and} \tag{14a}$$

$$\mathbb{E}\left[\ell_i(\phi_a(\mathbf{x}), y)\right] \leq au_i + (1-a)v_i, \quad \text{for } i = 1, \ldots, m. \tag{14b}$$

From (14b) we obtain that $\phi_a$ is feasible for $(\tilde{\mathsf{P}}_{\mathbf{u}_a})$. Hence, it is a suboptimal solution of $(\tilde{\mathsf{P}}_{\mathbf{u}_a})$, which implies that

$$\tilde{\mathsf{P}}^\star(\mathbf{u}_a) \leq \mathbb{E}\left[\ell_0(\phi_a(\mathbf{x}), y)\right] \leq a\tilde{\mathsf{P}}^\star(\mathbf{v}) + (1-a)\tilde{\mathsf{P}}^\star(\mathbf{w}), \tag{15}$$

thus concluding the proof. ∎

Proposition 5 shows that the perturbation function $\tilde{\mathsf{P}}^\star(\mathbf{u})$ can be convex even when $(\tilde{\mathsf{P}}_{\mathbf{u}})$ is defined based on non-convex losses. In the context of resilient learning, this is important because it allows us to quantify the variations of the perturbation function, which is the basis for the resilient equilibrium from Def. 1.

The convexity of the perturbation function $\tilde{\mathsf{P}}^\star(\mathbf{u})$ is also relevant because, under a constraint qualification such as the one stated in Assumption 2, it implies that the optimization problem is strongly dual. We encapsulate this result in the next lemma since it will be used in subsequent derivations.

**Lemma 2.** *Suppose $\tilde{\mathsf{P}}^\star(\mathbf{u})$ associated to $(\tilde{\mathsf{P}}_{\mathbf{u}})$ is convex (Assumption 1) and there exists $\phi \in \mathcal{F}$ strictly feasible for $(\tilde{\mathsf{P}}_{\mathbf{0}})$ (Assumption 2). Then, strong duality holds for $(\tilde{\mathsf{P}}_{\mathbf{u}})$, i.e., $\tilde{\mathsf{P}}^\star = \tilde{\mathsf{D}}^\star$.*

*Proof.* The result follows from [50, Thm. 15] since the constraint qualification in Assumption (2) implies that $\tilde{\mathsf{P}}^\star$ is lower semi-continuous [50, Thm. 18(a)], i.e., that its epigraph is closed [51, Prop. 1.1.2]. ∎

## C  Proof of Thm. 1

The proof is divided in three steps that transform the resilient functional dual problem $(\tilde{\mathsf{P}}\text{-RES})$ into the resilient, parametrized, empirical dual problem $(\hat{\mathsf{D}}\text{-RES})$. Explicitly, define the Lagrangian of $(\tilde{\mathsf{P}}\text{-RES})$ as

$$\tilde{L}_{\mathsf{R}}(\phi, \boldsymbol{\lambda}; \mathbf{u}) = \mathbb{E}_{\mathcal{D}_0}\left[\ell_0(\phi(\mathbf{x}), y)\right] + h(\mathbf{u}) + \sum_{i=1}^{m} \lambda_i \left(\mathbb{E}_{\mathcal{D}_i}\left[\ell_i(\phi(\mathbf{x}), y)\right] - u_i\right) \tag{16}$$

and its dual problem as

$$\tilde{\mathsf{D}}_{\mathsf{R}}^\star = \max_{\boldsymbol{\lambda} \in \mathbb{R}_+^m} \min_{\phi \in \mathcal{F}, \mathbf{u} \in \mathbb{R}_+^m} \tilde{L}_{\mathsf{R}}(\phi, \boldsymbol{\lambda}; \mathbf{u}). \tag{D-RES}$$

First, from lemma 2 we have that if assumption 1 and assumption 2 hold then the problem is strongly dual, i.e., $\tilde{\mathsf{P}}_{\mathsf{R}}^\star = \tilde{\mathsf{D}}_{\mathsf{R}}^\star$.

Then, to relate (D-RES) to its parametrized, empirical version $(\hat{\mathsf{D}}\text{-RES})$, we begin by approximating $\mathcal{F}$ by $\mathcal{F}_\theta$. Explicitly, we consider the difference between $\tilde{\mathsf{P}}_{\mathsf{R}}^\star$ and

$$\tilde{\mathsf{D}}_\theta^\star = \max_{\boldsymbol{\lambda} \in \mathbb{R}_+^m} \min_{\theta \in \Theta, \mathbf{u} \in \mathbb{R}_+^m} \tilde{L}_{\mathsf{R}}(f_\theta, \boldsymbol{\lambda}; \mathbf{u}). \tag{$D_\theta$-RES}$$

This difference, which we call the *approximation error*, is bounded in the following proposition:

**Proposition 6.** *Under Assumptions 1–2, it holds that*

$$\tilde{\mathsf{P}}_{\mathsf{R}}^\star \leq \tilde{\mathsf{D}}_\theta^\star \leq \tilde{\mathsf{P}}_{\mathsf{R}}^\star + h(\mathbf{u}^\star + \mathbb{1} \cdot M\nu) - h(\mathbf{u}^\star) + M\nu, \tag{17}$$

*for $\tilde{\mathsf{P}}_{\mathsf{R}}^\star$ and $\mathbf{u}^\star$ as in $(\tilde{\mathsf{P}}\text{-RES})$ and $\tilde{\mathsf{D}}_\theta^\star$ as in $(D_\theta\text{-RES})$.*

The proof of Prop. 6 is deferred to Sec. C.1. We proceed by replacing the expectations in $(D_\theta\text{-RES})$ by empirical averages, leading to a *generalization error* bounded as follows:

**Proposition 7.** *Let* $\tilde{\mathsf{D}}_\theta^\star$ *be the value of the parametrized dual problem* (D$_\theta$-RES) *achieved for a Lagrange multiplier* $\boldsymbol{\lambda}^\star$ *and* $\hat{\mathsf{D}}_R^\star$ *be the value of the empirical, parametrized dual problem* (D̂-RES) *achieved for a Lagrange multiplier* $\hat{\boldsymbol{\lambda}}^\star$. *Under Assumption* (5), *it holds that*

$$\left| \tilde{\mathsf{D}}_\theta^\star - \hat{\mathsf{D}}_R^\star \right| \le (1 + \Delta)\xi(N, \delta) \tag{18}$$

*with probability* $1 - 2(m + 1)\delta$, *where* $\Delta = \max\left( \|\boldsymbol{\lambda}^\star\|_1, \|\hat{\boldsymbol{\lambda}}^\star\|_1 \right)$.

A proof can be found in Sec. C.2. Combining the results in Prop. 6 and 7 using the triangle inequality yields (6) and concludes the proof of Thm. 1.

## C.1 Bounding the parametrization error: proof of Prop. 6

The lower bound stems directly from the definition of the dual problem in (D$_\theta$-RES). Indeed, it is immediate that

$$\tilde{\mathsf{D}}_\theta^\star \ge \min_{\theta \in \Theta, \, \mathbf{u} \in \mathbb{R}_+^m} \tilde{L}_R(f_\theta, \boldsymbol{\lambda}; \mathbf{u}), \quad \text{for all } \boldsymbol{\lambda} \in \mathbb{R}_+^m. \tag{19}$$

In particular, (19) holds for a solution $\boldsymbol{\lambda}^\star$ of the resilient functional dual problem D-RES. Explicitly, we can write

$$\tilde{\mathsf{D}}_\theta^\star \ge \min_{\theta \in \Theta, \, \mathbf{u} \in \mathbb{R}_+^m} \tilde{L}_R(f_\theta, \boldsymbol{\lambda}^\star; \mathbf{u})$$

Since $\mathcal{F}_\theta \subseteq \mathcal{F}$, it follows that

$$\tilde{\mathsf{D}}_\theta^\star \ge \min_{\theta \in \Theta, \, \mathbf{u} \in \mathbb{R}_+^m} \tilde{L}_R(f_\theta, \boldsymbol{\lambda}^\star; \mathbf{u}) \ge \min_{\phi \in \mathcal{F}, \, \mathbf{u} \in \mathbb{R}_+^m} \tilde{L}_R(\phi, \boldsymbol{\lambda}^\star; \mathbf{u}) = \tilde{\mathsf{D}}_R^\star = \tilde{\mathsf{P}}_R^\star. \tag{20}$$

The upper bound is obtained by relating the parameterized dual problem (D$_\theta$-RES) to a tightened version of (P̃-RES). Start by adding and subtracting $\tilde{L}_R(\phi, \boldsymbol{\lambda}; \mathbf{u})$ in (16) from the objective of the parametrized dual problem (D$_\theta$-RES) to get

$$\begin{aligned}
\tilde{L}_R(f_\theta, \boldsymbol{\lambda}; \mathbf{u}) &= \tilde{L}_R(\phi, \boldsymbol{\lambda}; \mathbf{u}) + \left[ \tilde{L}_R(f_\theta, \boldsymbol{\lambda}; \mathbf{u}) - \tilde{L}_R(\phi, \boldsymbol{\lambda}; \mathbf{u}) \right] \\
&= \tilde{L}_R(\phi, \boldsymbol{\lambda}; \mathbf{u}) + \mathbb{E}_{\mathcal{D}_0}\left[ \ell_0(f_\theta(\mathbf{x}), y) - \ell_0(\phi(\mathbf{x}), y) \right] \\
&\quad + \sum_{i=1}^m \lambda_i \mathbb{E}_{\mathcal{D}_i}\left[ \ell_i(f_\theta(\mathbf{x}), y) - \ell_i(\phi(\mathbf{x}), y) \right],
\end{aligned} \tag{21}$$

where the relaxations $u_i$ and relaxation costs $h(\mathbf{u})$ canceled out. To proceed, use the Lipschitz continuity of the losses (Assumption 3) to bound the expected loss difference as

$$\begin{aligned}
\left| \mathbb{E}_{\mathcal{D}_i}\left[ \ell_i\big(f_\theta(\mathbf{x}), y\big) - \ell_i(\phi(\mathbf{x}), y) \right] \right| &\le \mathbb{E}_{\mathcal{D}_i}\left[ \left| \ell_i\big(\phi(\mathbf{x}), y\big) - \ell_i\big(f_\theta(\mathbf{x}), y\big) \right| \right] \\
&\le M \mathbb{E}_{\mathcal{D}_i}\left[ |f_\theta - \phi(\mathbf{x})| \right], \quad \text{for } i = 0, \ldots, m.
\end{aligned}$$

We can then bound (21) as

$$\tilde{L}_R(f_\theta, \boldsymbol{\lambda}; \mathbf{u}) \le \tilde{L}_R(\phi, \boldsymbol{\lambda}; \mathbf{u}) + M\left( 1 + \sum_{i=1}^m \lambda_i \right) \left[ \max_{i \in \{0, \ldots, m\}} \mathbb{E}_{\mathcal{D}_i}\left[ |f_\theta - \phi(\mathbf{x})| \right] \right]. \tag{22}$$

Minimizing (22) over $\theta$ yields

$$\min_{\theta \in \Theta} \tilde{L}_R(f_\theta, \boldsymbol{\lambda}; \mathbf{u}) \le \tilde{L}_R(\phi, \boldsymbol{\lambda}; \mathbf{u}) + M\left( 1 + \sum_{i=1}^m \lambda_i \right) \left[ \min_{\theta \in \Theta} \max_{i \in \{0, \ldots, m\}} \mathbb{E}_{\mathcal{D}_i}\left[ |f_\theta - \phi(\mathbf{x})| \right] \right]. \tag{23}$$

However, from the $\nu$-approximation property of the parametrization (Assumption 4), for every $\phi \in \mathcal{F}$ there exists $\theta \in \Theta$ such that $\mathbb{E}_{\mathcal{D}_i}\left[ |f_\theta - \phi(\mathbf{x})| \right] \le \nu$, for $i = 0, \ldots, m$. Hence, (23) simplifies to

$$\min_{\theta \in \Theta} \tilde{L}_R(f_\theta, \boldsymbol{\lambda}; \mathbf{u}) \le \tilde{L}_R(\phi, \boldsymbol{\lambda}; \mathbf{u}) + \left( 1 + \sum_{i=1}^m \lambda_i \right) M\nu. \tag{24}$$

Notice that (24) holds uniformly over $\phi \in \mathcal{F}$. Additionally, since the left-hand side does not depend on $\phi$, minimizing it over $\mathbf{u}$ and $\phi$ then maximizing with respect to $\boldsymbol{\lambda}$ recover the dual value from (D$_\theta$-RES). We immediately obtain the upper bound

$$\tilde{\mathsf{D}}_\theta^\star \le \max_{\boldsymbol{\lambda} \in \mathbb{R}_+^m} \min_{\phi \in \mathcal{F}, \, \mathbf{u} \in \mathbb{R}_+^m} \tilde{L}_R(\phi, \boldsymbol{\lambda}; \mathbf{u}) + \left( 1 + \sum_{i=1}^m \lambda_i \right) M\nu. \tag{25}$$

Expanding $\tilde{L}_R$ and rearranging the terms in (25), we recognize the Lagrangian of a perturbed version of (P̃-RES), namely

$$\tilde{P}^{\star}_{M\nu} = \min_{\phi\in\mathcal{F},\,\mathbf{u}\in\mathbb{R}^m_+} \quad \mathbb{E}_{\mathcal{D}_0}\Big[\ell_0\big(\phi(\mathbf{x}),y\big)\Big] + h(\mathbf{u}) + M\nu$$

$$\text{subject to} \quad \mathbb{E}_{(\mathbf{x},y)\sim\mathcal{D}_i}\Big[\ell_i\big(\phi(\mathbf{x}),y\big)\Big] \leq u_i - M\nu, \quad i = 1,\ldots,m. \tag{PVI}$$

Hence, the right-hand side of (25) is the dual problem of (PVI). Since (PVI) is strongly dual (from lemma 2), we obtain that $\tilde{D}^{\star}_\theta \leq \tilde{P}^{\star}_{M\nu}$. To bound $\tilde{P}^{\star}_{M\nu}$, note that we can construct a feasible pair for (PVI) from any pair $(\phi^\star, \mathbf{u}^\star)$ that solves the resilient functional learning problem (P̃-RES). Explicitly, observe that the pair $(\phi^\star, \mathbf{u}^\star + \mathbb{1}\cdot M\nu)$ is a (suboptimal) feasible point of (PVI). Thus,

$$\tilde{D}^{\star}_\theta \leq \tilde{P}^{\star}_{M\nu} \leq \mathbb{E}_{\mathcal{D}_0}\Big[\ell_0\big(\phi^\star(\mathbf{x}),y\big)\Big] + h(\mathbf{u}^\star + \mathbb{1}\cdot M\nu) + M\nu$$

$$\leq \tilde{P}^{\star}_R - h(\mathbf{u}^\star) + h(\mathbf{u}^\star + \mathbb{1}\cdot M\nu) + M\nu \tag{26}$$

Combining (20) and (26) concludes the proof. ∎

### C.2   Bounding the generalization error: Proof of Prop. 7

Let $\boldsymbol{\lambda}^\star$ and $\hat{\boldsymbol{\lambda}}^\star$ be solution pairs of (D$_\theta$-RES) and (D̂-RES) respectively and consider the sets of dual minimizers

$$\mathcal{M}(\boldsymbol{\lambda}) = \operatorname*{argmin}_{\theta\in\Theta,\,\mathbf{u}\in\mathbb{R}^m_+} \tilde{L}_R(f_\theta, \boldsymbol{\lambda}; \mathbf{u}) \quad \text{and} \quad \hat{\mathcal{M}}(\boldsymbol{\lambda}) = \operatorname*{argmin}_{\theta\in\Theta,\,\mathbf{u}\in\mathbb{R}^m_+} \hat{L}_\theta(\theta, \boldsymbol{\lambda}; \mathbf{u}).$$

Note that member of these set are pairs $(\theta, \mathbf{u})$. Since $\hat{\boldsymbol{\lambda}}^\star$ maximizes the minimum (with respect to $\theta$ and $\mathbf{u}$) of the empirical Lagrangian in (5), it holds that $\min_{\theta\in\Theta,\,\mathbf{u}\in\mathbb{R}^m_+} \hat{L}_\theta(\theta, \hat{\boldsymbol{\lambda}}^\star; \mathbf{u}) \geq \min_{\theta\in\Theta,\,\mathbf{u}\in\mathbb{R}^m_+} \hat{L}_\theta(\theta, \boldsymbol{\lambda}; \mathbf{u})$ for all $\boldsymbol{\lambda}\in\mathbb{R}^m_+$. In particular,

$$\tilde{D}^{\star}_\theta - \hat{D}^{\star}_R = \min_{\theta\in\Theta,\,\mathbf{u}\in\mathbb{R}^m_+} \tilde{L}_R(f_\theta, \boldsymbol{\lambda}^\star; \mathbf{u}) - \min_{\theta\in\Theta,\,\mathbf{u}\in\mathbb{R}^m_+} \hat{L}_\theta(\theta, \hat{\boldsymbol{\lambda}}^\star; \mathbf{u})$$

$$\leq \min_{\theta\in\Theta,\,\mathbf{u}\in\mathbb{R}^m_+} \tilde{L}_R(f_\theta, \boldsymbol{\lambda}^\star; \mathbf{u}) - \min_{\theta\in\Theta,\,\mathbf{u}\in\mathbb{R}^m_+} \hat{L}_\theta(\theta, \boldsymbol{\lambda}^\star; \mathbf{u}).$$

Since $(\hat{\theta}^\dagger, \hat{\mathbf{u}}^\dagger) \in \hat{\mathcal{M}}(\boldsymbol{\lambda}^\star)$ is suboptimal for $\tilde{L}_R(f_\theta, \boldsymbol{\lambda}^\star; \mathbf{u})$, we get

$$\tilde{D}^{\star}_\theta - \hat{D}^{\star}_R \leq \tilde{L}_R(f_{\hat{\theta}^\dagger}, \boldsymbol{\lambda}^\star; \hat{\mathbf{u}}^\dagger) - \hat{L}_\theta(\hat{\theta}^\dagger, \boldsymbol{\lambda}^\star; \hat{\mathbf{u}}^\dagger).$$

Using a similar argument yields

$$\tilde{D}^{\star}_\theta - \hat{D}^{\star}_R \geq \tilde{L}_R(f_{\theta^\dagger}, \hat{\boldsymbol{\lambda}}^\star; \mathbf{u}^\dagger) - \hat{L}_\theta(\theta^\dagger, \hat{\boldsymbol{\lambda}}^\star; \mathbf{u}^\dagger)$$

for $(\theta^\dagger, \mathbf{u}^\dagger) \in \mathcal{M}(\hat{\boldsymbol{\lambda}}^\star)$. We thus obtain

$$\Big|\tilde{D}^{\star}_\theta - \hat{D}^{\star}_R\Big| \leq \max\left\{\Big|\tilde{L}_R(f_{\hat{\theta}^\dagger}, \boldsymbol{\lambda}^\star; \hat{\mathbf{u}}^\dagger) - \hat{L}_\theta(\hat{\theta}^\dagger, \boldsymbol{\lambda}^\star; \hat{\mathbf{u}}^\dagger)\Big|, \Big|\tilde{L}_R(f_{\theta^\dagger}, \hat{\boldsymbol{\lambda}}^\star; \mathbf{u}^\dagger) - \hat{L}_\theta(\theta^\dagger, \hat{\boldsymbol{\lambda}}^\star; \mathbf{u}^\dagger)\Big|\right\}$$

$$\tag{27}$$

Using the uniform convergence bound from Assumption 5, we obtain that

$$\Big|\tilde{L}_R(f_\theta, \boldsymbol{\lambda}; \mathbf{u}) - \hat{L}_\theta(\theta, \boldsymbol{\lambda}; \mathbf{u})\Big| \leq \xi(N, \delta) + \sum_{i=1}^m \lambda_i \xi(N, \delta) \leq (1 + \|\boldsymbol{\lambda}\|_1)\xi(N, \delta), \tag{28}$$

holds uniformly over $\theta$ with probability $1 - (m+1)\delta$. Combining (27) with (28) using the union bound concludes the proof. ∎

## D   Proofs from Main Text

### D.1   Proof of Proposition 1

*Proof.* **[Existence]** To show there exists $\mathbf{u}^\star \in \mathbb{R}^m_+$ satisfying Def. 1, notice that (2) is equivalent to showing that there exists $\mathbf{u}^\star$ such that $\mathbf{0} \in \partial f(\mathbf{u}^\star)$ for $f(\mathbf{u}) = \tilde{P}^\star(\mathbf{u}) + h(\mathbf{u})$. Notice that $f$ is a convex function, so that its subgradient is well-defined [see (1)].

Start by noticing that the set of minimizers of $f$ is not empty. Explicitly, let $\mathcal{U}^\star = \text{argmin}_{\mathbf{u} \in \mathbb{R}_+^m} f(\mathbf{u})$. Observe that since the losses $\ell_i$ are bounded, $\tilde{\mathsf{P}}^\star(\bar{\mathbf{u}}) = \tilde{\mathsf{P}}^\star(\mathbf{v})$ for all $\mathbf{v} \succeq \bar{\mathbf{u}} = B \cdot \mathbb{1}$. Immediately, we obtain that $f(\mathbf{v})$ is a componentwise increasing function for $\mathbf{v} \succeq \bar{\mathbf{u}}$, which implies that $\mathcal{U}^\star = \text{argmin}_{\mathbf{u} \in \bar{\mathcal{U}}} f(\mathbf{u})$, where $\bar{\mathcal{U}} = \{\mathbf{u} \in \mathbb{R}_+^m \mid \mathbf{u} \preceq \bar{\mathbf{u}}\}$. Since $\bar{\mathcal{U}}$ is compact and $f$ is convex, its set of minimizers $\mathcal{U}^\star$ is compact and non-empty [51, Prop. B.10(b)].

For all $\mathbf{u}' \in \text{int}(\mathcal{U}^\star)$, where $\text{int}(\mathcal{Z})$ denotes the interior of the set $\mathcal{Z}$, it must be that $\mathbf{0} \in \partial f(\mathbf{u}')$ [52, Prop. 5.4.7]. This is simply a statement of the first-order optimality condition for $\mathbf{u}'$.

Suppose, now, that $\mathcal{U}^\star$ has no interior and that for all $\mathbf{u}' \in \mathcal{U}^\star$, $[\mathbf{u}']_j = 0$ for some $j$. Then, it could be that for all $\mathbf{p} \in \partial f(\mathbf{u}')$ are such that $[\mathbf{p}]_j > 0$. However, this would lead to a contradiction. Indeed, since $h$ is normalized, it holds that $[\nabla h(\mathbf{u}')]_j = 0$, so that $\partial f(\mathbf{u}') = \partial \tilde{\mathsf{P}}^\star(\mathbf{u}')$. Thus, if all $\mathbf{p} \in \partial f(\mathbf{u}')$ are such that $[\mathbf{p}]_j > 0$, then all $\mathbf{p} \in \partial \tilde{\mathsf{P}}^\star(\mathbf{u}')$ are such that $[\mathbf{p}]_j > 0$. From the convexity of $\tilde{\mathsf{P}}^\star$, this would imply that $\tilde{\mathsf{P}}^\star(\mathbf{u}' + \mathbf{e}_j) \geq \tilde{\mathsf{P}}^\star(\mathbf{u}') + [\mathbf{p}]_j > \tilde{\mathsf{P}}^\star(\mathbf{u}')$, where $\mathbf{e}_j$ is the $j$-th element of the canonical basis. This contradicts the fact that $\tilde{\mathsf{P}}^\star$ is componentwise non-increasing. It must therefore be that $\mathbf{0} \in \partial f(\mathbf{u}')$.

**[Unicity]** If $h$ is strictly convex, then $f$ is strictly convex. Then, its minimizer is unique [53, Chap. 27], i.e., $\mathcal{U}^\star$ is a singleton. Suppose there exists $\mathbf{u}^\dagger \notin \mathcal{U}^\star$ that satisfies the equilibrium in Def. 1. Then, from (2), it holds that $\mathbf{0} \in \partial f(\mathbf{u}^\dagger)$. Since $f$ is convex, this implies that $\mathbf{u}^\dagger$ is a minimizer, which violates the hypothesis that it is not in $\mathcal{U}^\star$. ∎

### D.2 Proposition 2

*Proof.* From the definition of sub-differential, it holds for any $\mathbf{p}_v \in \partial \tilde{\mathsf{P}}^\star(\mathbf{v})$ and $\mathbf{p}_w \in \partial \tilde{\mathsf{P}}^\star(\mathbf{w})$ that

$$\tilde{\mathsf{P}}^\star(\mathbf{v}) \geq \tilde{\mathsf{P}}^\star(\mathbf{w}) + \mathbf{p}_w^T(\mathbf{v} - \mathbf{w}) \tag{29}$$

$$\tilde{\mathsf{P}}^\star(\mathbf{w}) \geq \tilde{\mathsf{P}}^\star(\mathbf{v}) + \mathbf{p}_v^T(\mathbf{w} - \mathbf{v}) \tag{30}$$

Adding (29) and (30), we can rearrange the terms to get

$$0 \geq (\mathbf{p}_v - \mathbf{p}_w)^T(\mathbf{w} - \mathbf{v}) \tag{31}$$

Then, since $\mathbf{v}_i - \mathbf{u}_i > 0$ if $i = j$ and 0 otherwise, (31) implies that $[\mathbf{p}_v]_j - [\mathbf{p}_w]_j \leq 0$, which yields the desired result. Applying the same argument for $h$ concludes the proof. ∎

### D.3 Proposition 3

*Proof.* The strong duality of ($\tilde{\mathsf{P}}$-RES) under assumptions 1 and 2 holds due to Lemma 2. The proof then follows the same steps used to prove that optimal Lagrange multipliers are subgradients of the dual function in convex problems; see, e.g., [39, Section 5.6.2]. We just verify that the proof holds despite the non-convexity of the problem as long as the problem is strongly dual.

By definition of the dual function we can write

$$\tilde{\mathsf{P}}^\star(\mathbf{u}) = \mathsf{D}^\star(\mathbf{u}) = g(\boldsymbol{\lambda}^*; \mathbf{u}) = \min_\phi \mathcal{L}\left(\phi, \boldsymbol{\lambda}^*(\mathbf{u}); \mathbf{u}\right) \leq \mathcal{L}\left(\phi, \boldsymbol{\lambda}^*(\mathbf{u}); \mathbf{u}\right) \tag{32}$$

where the inequality is true for any function $\phi$. We particularize this inequality to a function $\phi^*(\cdot; \mathbf{v})$ that attains the minimum of ($\mathsf{P}_\mathbf{u}$) for relaxation $\mathbf{v}$. We can therefore write

$$\tilde{\mathsf{P}}^\star(\mathbf{u}) \leq \mathcal{L}\left(\phi^*(\cdot; \mathbf{v}), \boldsymbol{\lambda}^*(\mathbf{u}); \mathbf{u}\right), \tag{33}$$

We now substitute the definition of the Lagrangian in (3) for the right hand side of (33) to write

$$\tilde{\mathsf{P}}^\star(\mathbf{u}) \leq \mathbb{E}_{\mathcal{D}_0}\left[\ell_0(\phi^*(\mathbf{x}; \mathbf{v}), y)\right] + \sum_{i=1}^m \lambda_i^*(\mathbf{u})\left[\mathbb{E}_{\mathcal{D}_i}\left[\ell_i(\phi^*(\mathbf{x}; \mathbf{v}), y)\right] - u_i\right]. \tag{34}$$

The important property to observe in (34) is that we consider perturbation $\mathbf{u}$ and corresponding dual variable $\boldsymbol{\lambda}^*(\mathbf{u})$ while evaluating the Lagrangian at the function $\phi^*(\mathbf{x}; \mathbf{v})$ that is primal optimal for perturbation $\mathbf{v}$. This latter fact implies that the following equality and inequalities are true

$$\mathbb{E}_{\mathcal{D}_0}\left[\ell_0(\phi^*(\mathbf{x}; \mathbf{v}), y)\right] = \tilde{\mathsf{P}}^\star(\mathbf{v}), \quad \mathbb{E}_{\mathcal{D}_i}\left[\ell_i(\phi^*(\mathbf{x}; \mathbf{v}), y)\right] \leq -v_i. \tag{35}$$

Using the relationships in (35) and (34) we conclude that

$$\tilde{\mathsf{P}}^\star(\mathbf{u}) \leq \tilde{\mathsf{P}}^\star(\mathbf{v}) + \sum_{i=1}^m \lambda_i^*(\mathbf{u})\Big[v_i - u_i\Big]. \tag{36}$$

The two sums in the right hand side of (36) equal the inner product of multiplier $\boldsymbol{\lambda}^\star(\mathbf{u})$ with $(\mathbf{v} - \mathbf{u})$. Implementing this substitution in (36) and reordering terms yields

$$\tilde{\mathsf{P}}^\star(\mathbf{v}) \geq \tilde{\mathsf{P}}^\star(\mathbf{u}) - \boldsymbol{\lambda}^*(\mathbf{u})^T(\mathbf{v} - \mathbf{u}). \tag{37}$$

Comparing (37) with (1) we see that $\boldsymbol{\lambda}^*(\mathbf{u})$ satisfies Definition 1. ∎

## D.4 Proposition 4

*Proof.* As we did in the proof of Prop. 1, we begin by showing that $\mathbf{u}^\star$ is a minimizer of $(\tilde{\mathsf{P}}^\star + h)(\mathbf{u}) = \tilde{\mathsf{P}}^\star(\mathbf{u}) + h(\mathbf{u})$. Explicitly,

$$\mathbf{u}^\star \in \operatorname*{argmin}_{\mathbf{u} \in R_+^m} (\tilde{\mathsf{P}}^\star + h)(\mathbf{u}). \tag{$\tilde{\mathsf{P}}$-RES$'$}$$

As shown in proposition 1, $\mathbf{u}^\star$ minimizes $\tilde{\mathsf{P}}^\star + h$ iff $0 \in \partial(\tilde{\mathsf{P}}^\star + h)(\mathbf{u}^\star)$. Then $0 \in \partial(\tilde{\mathsf{P}}^\star + h)(\mathbf{u}^\star)$ is also equivalent to the resilient equilibrium definition (2).

In $(\tilde{\mathsf{P}}_\mathbf{u})$, the function $\tilde{\mathsf{P}}^\star(\mathbf{u})$ is defined as the solution of $(\tilde{\mathsf{P}}_\mathbf{u})$. To show that $(\tilde{\mathsf{P}}$-RES$')$ and $(\tilde{\mathsf{P}}$-RES$)$ are equivalent a nested minimization over variables $\phi$ and $\mathbf{u}$ is equivalent to a joint minimization over $\phi$ and $\mathbf{u}$. To confirm that this holds here recall the definition of $\mathbf{u}^\star$ as the resilient relaxation and of $\phi^*(\mathbf{x}; \mathbf{u})$ as the minimizer of the relaxed problem $(\tilde{\mathsf{P}}_\mathbf{u})$ – which holds for all $\mathbf{u}$ and $\mathbf{u}^\star$ in particular. Since $\mathbf{u}^\star$ is the solution of $(\tilde{\mathsf{P}}$-RES$')$. We therefore have that for all perturbations $\mathbf{u}$

$$\mathbb{E}_{(\mathbf{x},y)\sim\mathcal{D}_0}\Big[\ell_0(\phi^*(\mathbf{x};\mathbf{u}^\star),y)\Big] + h(\mathbf{u}^\star) \leq \mathbb{E}_{(\mathbf{x},y)\sim\mathcal{D}_0}\Big[\ell_0(\phi^*(\mathbf{x};\mathbf{u}),y)\Big] + h(\mathbf{u}). \tag{38}$$

Further observe that $\phi^*(\mathbf{x}; \mathbf{u})$ is the minimizer of the relaxed problem $(\mathsf{P}_\mathbf{u})$ associated with perturbation $\mathbf{u}$. We then have that for any feasible function $\phi$ we must have

$$\mathbb{E}_{(\mathbf{x},y)\sim\mathcal{D}_0}\Big[\ell_0(\phi^*(\mathbf{x};\mathbf{u}),y)\Big] + h(\mathbf{u}) \leq \mathbb{E}_{(\mathbf{x},y)\sim\mathcal{D}_0}\Big[\ell_0(\phi(\mathbf{x}),y)\Big] + h(\mathbf{u}). \tag{39}$$

The bound in (39) is true for all $\phi$ and $\mathbf{u}$. In particular, it is true for the solution $\phi^*, \mathbf{u}^*$ of $(\tilde{\mathsf{P}}$-RES$)$. Particularizing (39) to this pair and combining the resulting inequality with the bound in (38) we conclude that

$$\mathbb{E}_{(\mathbf{x},y)\sim\mathcal{D}_0}\Big[\ell_0(\phi^*(\mathbf{x};\mathbf{u}^\star),y)\Big] + h(\mathbf{u}^\star) \leq \mathbb{E}_{(\mathbf{x},y)\sim\mathcal{D}_0}\Big[\ell_0(\phi^*(\mathbf{x}),y)\Big] + h(\mathbf{u}^*). \tag{40}$$

On the other hand, given that $\phi^*, \mathbf{u}^*$ solve $(\tilde{\mathsf{P}}$-RES$)$ we have that for all feasible $\mathbf{u}$ and $\phi$

$$\mathbb{E}_{(\mathbf{x},y)\sim\mathcal{D}_0}\Big[\ell_0(\phi^*(\mathbf{x}),y)\Big] + h(\mathbf{u}^*) \leq \mathbb{E}_{(\mathbf{x},y)\sim\mathcal{D}_0}\Big[\ell_0(\phi(\mathbf{x}),y)\Big] + h(\mathbf{u}). \tag{41}$$

In particular, this is true if we make $\mathbf{u} = \mathbf{u}^\star$ and $\phi = \phi^*(\mathbf{x}; \mathbf{u}^\star)$. We can then write,

$$\mathbb{E}_{(\mathbf{x},y)\sim\mathcal{D}_0}\Big[\ell_0(\phi^*(\mathbf{x}),y)\Big] + h(\mathbf{u}^*) \leq \mathbb{E}_{(\mathbf{x},y)\sim\mathcal{D}_0}\Big[\ell_0(\phi^*(\mathbf{x};\mathbf{u}^\star),y)\Big] + h(\mathbf{u}^\star) \tag{42}$$

For (40) and (42) to hold we must have that $\phi^*, \mathbf{u}^*$ is a solution of $(\tilde{\mathsf{P}}$-RES$)$ if and only if $\mathbf{u}^\star$ and $\phi = \phi^*(\mathbf{x}; \mathbf{u}^\star)$ are a resilient perturbation and a corresponding resilient minimizer. ∎

## D.5 Proposition 8

**Proposition 8.** *Let* $\Phi^\star = \operatorname{argmin}_{\phi\in\mathcal{F}} \mathbb{E}_{(\mathbf{x},y)\sim\mathcal{D}_0}\Big[\ell_0(\phi(\mathbf{x};\mathbf{u}^\star),y)\Big]$ *denote the solution set of the unconstrained problem and* $\mathbf{b}_i = \min_{\phi\in\Phi^\star} \mathbb{E}_{(\mathbf{x},y)\sim\mathcal{D}_i}\Big[\ell_i(\phi(\mathbf{x};\mathbf{u}^\star),y)\Big]$, $i = 1,\ldots,m$ *be the minimum constraint losses achieved. We show there exists costs such that* $\underline{u}_i^\star \leq \epsilon$ *and* $\bar{u}_i^\star \geq b_i - \epsilon$ *for* $i = 1,\ldots,m$.

*Proof.*

First, we will show sufficient conditions for $\mathbf{u}^\star \preceq \mathbb{1}\epsilon$ and $\mathbf{u}^\star \succeq \mathbf{b} - \mathbb{1}\epsilon$, in terms of $\partial \tilde{\mathsf{P}}^\star$ and $\nabla h$ evaluated at $\mathbb{1}\epsilon$ and $\mathbf{b} - \mathbb{1}\epsilon$, respectively. For any $\mathbf{u} \in \mathbb{R}_+^m$ the strong duality of $\tilde{\mathsf{P}}^\star$ (lemma 2) implies that $\partial \tilde{\mathsf{P}}^\star(\mathbf{u})$ is non empty and bounded. Also, the convexity of $\tilde{\mathsf{P}}^\star$ (assumption 1) implies that for all $\mathbf{p_u} \in \partial \tilde{\mathsf{P}}^\star(\mathbf{u})$ and $\mathbf{p_{u^\star}} \in \partial \tilde{\mathsf{P}}^\star(\mathbf{u}^\star)$ by (31) from proposition (3) it holds that

$$\left\langle \mathbf{p_{u^\star}} - \mathbf{p_u}, \mathbf{u}^\star - \mathbf{u} \right\rangle \geq 0 \tag{43}$$

Analogously, due to the convexity of $h$ we have that

$$\left\langle \nabla h(\mathbf{u}^\star) - \nabla h(\mathbf{u}), \mathbf{u}^\star - \mathbf{u} \right\rangle \geq 0 \tag{44}$$

Adding (43) and (44) we obtain

$$\left\langle \left[ \mathbf{p_{u^\star}} + \nabla h(\mathbf{u}^\star) \right] - \left[ \mathbf{p_u} + \nabla h(\mathbf{u}) \right], \mathbf{u}^\star - \mathbf{u} \right\rangle \geq 0 \tag{45}$$

(45) holds for any $\mathbf{p_{u^\star}} \in \partial \tilde{\mathsf{P}}^\star(\mathbf{u}^\star)$. Since $\mathbf{u}^\star$ achieves the resilient equilibrium, by definition $-\nabla h(\mathbf{u}^\star) = \mathbf{p_{u^\star}} \in \partial \tilde{\mathsf{P}}^\star(\mathbf{u}^\star)$, which implies that for all $\mathbf{p_u} \in \partial \tilde{\mathsf{P}}^\star(\mathbf{u})$

$$\left\langle \left[ \mathbf{p_u} + \nabla h(\mathbf{u}) \right], \mathbf{u}^\star - \mathbf{u} \right\rangle \leq 0 \tag{46}$$

Finally, by choosing particular values of $\mathbf{u}$ in (46) we obtain the desired conditions.

$(\mathbf{u}^\star \preceq \mathbb{1}\epsilon)$: Letting $a = \epsilon$, (46) implies that if there exists

$$\mathbf{p}_{\mathbb{1}\epsilon} \in \partial \tilde{\mathsf{P}}^\star(\mathbb{1}\epsilon) \text{ such that } \mathbf{p}_{(\mathbb{1}\epsilon)} + h(\mathbb{1}\epsilon) \succ 0 \text{ then } \mathbf{u}^\star \preceq \mathbb{1}\epsilon. \tag{47}$$

$(\mathbf{u}^\star \succeq \mathbb{1}\mathbf{b} - \mathbb{1}\epsilon)$: Letting $a = \mathbf{b} - \mathbb{1}\epsilon$, (46) implies that if there exists

$$\mathbf{p}_{(\mathbf{b}-\mathbb{1}\epsilon)_+} \in \partial \tilde{\mathsf{P}}^\star((\mathbf{b} - \mathbb{1}\epsilon)_+) \text{ such that } \mathbf{p}_{(\mathbf{b}-\mathbb{1}\epsilon)_+} + h((\mathbf{b} - \mathbb{1}\epsilon)_+) \prec 0 \text{ then } \mathbf{u}^\star \succeq (\mathbf{b} - \mathbb{1}\epsilon)_+. \tag{48}$$

Let $h(\mathbf{u}) = \alpha \|\mathbf{u}\|_2^2$, $\alpha \in \mathbb{R}_+$, which has $\nabla h(\mathbf{u}) = 2\alpha \mathbf{u}$. We will now show that there exist coefficients $\bar{\alpha}$ and $\underline{\alpha}$ such that the associated costs $\bar{h}$ and $\underline{h}$ satisfy the conditions (47) and (48).

$(\underline{h})$: Let $\underline{\alpha} > \frac{\|\mathbf{p}_{\mathbb{1}\epsilon}\|_\infty}{2\epsilon}$, then

$$\nabla \underline{h}(\mathbb{1}\epsilon)_i = 2\underline{\alpha}\epsilon > [-\mathbf{p}_{\mathbb{1}\epsilon}]_i \text{ for all } i = 1, \ldots, m$$

which implies that $\underline{h}$ satisfies (47) and thus $\underline{\mathbf{u}}^\star \preceq \mathbb{1}\epsilon$.

$(\bar{h})$: Let $\bar{\alpha} < \min_i \frac{[-\mathbf{P}_{(\mathbf{b}-\mathbb{1}\epsilon)_+}]_i}{2(\mathbf{b}_i - \epsilon)}$, then

$$\nabla \bar{h}((\mathbf{b} - \mathbb{1}\epsilon)_+)_i = 2\bar{\alpha}(\mathbf{u}_i - \epsilon) < [\mathbf{p}_{\mathbb{1}(\mathbf{b}_i - \epsilon)}]_i \text{ for all } i = 1, \ldots, m$$

which implies that $\bar{h}$ satisfies (48) and thus $\bar{\mathbf{u}}^\star \succeq \mathbf{b} - \mathbb{1}\epsilon$. Since the definition of $\mathbf{b}$ and $\epsilon$ ensures that $\min_i \frac{[-\mathbf{P}_{(\mathbf{b}-\mathbb{1}\epsilon)_+}]_i}{2(\mathbf{b}_i - \epsilon)} > 0$, there exists $\bar{\alpha} > 0$ that satisfies the above condition. ∎

## E  Primal and Dual Updates

In section 4 we introduced the unconstrained problem ($\hat{\mathsf{D}}$-RES) that approximates our original problem ($\mathsf{P_u}$) as the saddle point of the empirical Lagrangian $\hat{L}_\theta$:

$$\hat{L}_\theta(\theta, \boldsymbol{\lambda}; \mathbf{u}) = h(\mathbf{u}) + \frac{1}{N} \sum_{n=1}^N \ell_0\big(f_\theta(\mathbf{x}_{n,0}), y_{n,0}\big) + \sum_{i=1}^m \lambda_i \left( \frac{1}{N} \sum_{n=1}^N \ell_i\big(f_\theta(\mathbf{x}_{n,i}), y_{n,i}\big) - u_i \right). \tag{49}$$

As described in Algorithm 1 he saddle point can be obtained by alternating the minimization and the maximization of the Lagrangian with respect to the primal variables and dual variables respectively. We now describe the updates of both primal and dual variables in greater detail. For clarity, we

describe the updates using gradient descent and sub-gradient ascent for primal and dual variables, respectively. However, nothing precludes our method from using other optimization algorithms, such as Adam [54] or SGD with nesterov momentum.

The optimization is done via gradient descent and $\mathbf{u}$ and stochastic gradient descent for the primal variables $\theta$

$$\theta^{(t+1)} = \theta^{(t)} - \eta_\theta \hat{d}\theta^{(t)}$$
$$\mathbf{u}^{(t+1)} = \mathbf{u}^{(t)} - \eta_\mathbf{u} d\mathbf{u}^{(t)}, \tag{50}$$

and stochastic subgradient ascent for the dual variables

$$\boldsymbol{\lambda}^{(t+1)} = \boldsymbol{\lambda}^{(t)} + \eta_\lambda \hat{d}\boldsymbol{\lambda}^{(t)}. \tag{51}$$

In order to obtain the gradients of the Lagrangian with respect to the primal variables, observe that the minimization of the Lagrangian can be separated into two parts that each depend on only one primal variable.

$$\hat{\mathcal{L}}(\theta, \boldsymbol{\lambda}, \mathbf{u}) = \hat{\mathcal{L}}_\theta(\theta, \boldsymbol{\lambda}) + \hat{\mathcal{L}}_\mathbf{u}(\boldsymbol{\lambda}, \mathbf{u}) \tag{52}$$

where $\hat{\mathcal{L}}_\phi(\phi, \boldsymbol{\lambda}) = \hat{\mathcal{L}}(\phi, \boldsymbol{\lambda}; 0)$ and $\hat{\mathcal{L}}_\mathbf{u}(\boldsymbol{\lambda}, \mathbf{u})$ is

$$\hat{\mathcal{L}}_\mathbf{u}(\boldsymbol{\lambda}, \mathbf{u}) = h(\mathbf{u}) - \boldsymbol{\lambda}^\top \mathbf{u}. \tag{53}$$

The gradient with respect to $\mathbf{u}$ is obtained from (53)

$$d\mathbf{u} = \frac{\partial \hat{\mathcal{L}}(\boldsymbol{\lambda}, \mathbf{u})}{\partial \mathbf{u}} = \nabla h(\mathbf{u}) - \boldsymbol{\lambda}. \tag{54}$$

Solving for the optimal $\theta$ is similar to solving a regularized empirical risk minimization problem. The gradient with respect to $\theta$ is estimated from $\hat{\mathcal{L}}(\theta, \boldsymbol{\lambda}; 0)$ using a minibatch of $B$ samples as

$$\hat{d}\theta = \frac{1}{B} \sum_{n=1}^{B} \left( \nabla_\theta \ell_0(f_\theta(\mathbf{x}_n), y_n) + \sum_{i=1}^{m} \nabla_\theta \ell_i(f_\theta(\mathbf{x}_n), y_n) \right). \tag{55}$$

The constraints evaluated at the optimal Lagrangian minimizers are supergradients of the corresponding Lagrange multipliers [55]. In the same manner, stochastic supergradient ascent is done by updating the dual variable $\boldsymbol{\lambda}$ with the super-gradient estimated using a batch of $B$ samples.

$$\hat{d}\boldsymbol{\lambda}_i = \frac{1}{B} \sum_{n=1}^{B} \ell_i(f_\theta(\mathbf{x}_n), y_n) - \mathbf{u}_i \; i = 1 \ldots m. \tag{56}$$

## F  Heterogenous Federated Learning Experiments

### F.1  Formulation

In this section, we show that the constrained problem $P$-FL can be re-written as a constrained learning problem of the form of P, and then introduce its resilient version,

Let $\mathfrak{D}_i$ be the distribution of data pairs for Client $i$, and $R_i(f_\theta) = \mathbb{E}_{(\mathbf{x},y) \sim \mathfrak{D}_i}[\ell(f_\theta(\mathbf{x}), y)]$ its statistical risk. We denote the average performance as $\overline{R}(f_\theta) := (1/C) \sum_{i=1}^{C} R_i(f_\theta)$, where $C$ is the number of clients. As proposed in [18] heterogeneity issues in federated learning can be tackled by imposing a proximity constraint between the performance of each client $R_i$, and the loss averaged over all clients $\overline{R}$. This leads to the constrained learning problem:

$$\min_{\theta \in \Theta} . \quad \overline{R}(f_\theta) \tag{P-FL}$$
$$\text{s. to} \quad R_i(f_\theta) - \overline{R}(f_\theta) - \epsilon \leq 0, \qquad i = 1, \ldots, C,$$

where $C$ is the number of clients and $\mathfrak{D}_i$ is the distribution of data pairs for client $i$, and $\epsilon$ is a small (fixed) positive scalar.

In order to show that problem $P$-FL can be re-written as a constrained learning problem of the form of P, we introduce a random variable $c$ uniformly distributed over client indices $\{1, \ldots, C\}$, i.e.,

$$P(c = i) = \frac{1}{C} \text{ for all } i = 1, \ldots, C$$

Then, we construct a mixture distribution $\overline{\mathfrak{D}}$ for data pairs such that

$$(\mathbf{x}, y)|c = i \sim \mathfrak{D}_i.$$

Then we can re-write the average risk $R(f_{\boldsymbol{\theta}})$ as the expected loss under $\overline{\mathfrak{D}}$:

$$\mathbb{E}_{\overline{\mathfrak{D}}}\left[\ell(f_{\boldsymbol{\theta}}(\mathbf{x}), y)\right] = \sum_{i=1}^{C} \frac{1}{C} \mathbb{E}_{\mathfrak{D}_i}\left[\ell(f_{\boldsymbol{\theta}}(\mathbf{x}), y)\right]$$
$$= R_i(f_{\boldsymbol{\theta}}),$$

where the first inequality holds by law of total expectation.

Let $\ell_i(f_{\boldsymbol{\theta}}(\mathbf{x}), y) = (C\mathbb{1}(c = i) - 1)\ell(f_{\boldsymbol{\theta}}(\mathbf{x}), y) - \epsilon$, where $\mathbb{1}(c = i)$ is the indicator function for client $i$.

In the same manner, we can re-write the $i - th$ constraint $R_i(f_{\boldsymbol{\theta}}) - \overline{R}(f_{\boldsymbol{\theta}}) - \epsilon$ as the expectation of the loss $\ell_i$ under $\overline{\mathfrak{D}}_i$:

$$\mathbb{E}_{\overline{\mathfrak{D}}}\left[\ell_i(f_{\boldsymbol{\theta}}(\mathbf{x}), y)\right] = \mathbb{E}_{\mathfrak{D}_i}\left[\ell(f_{\boldsymbol{\theta}}(\mathbf{x}), y)\right] - \epsilon - \sum_{j=1}^{C} \frac{1}{C} \mathbb{E}_{\mathfrak{D}_j}\left[\ell(f_{\boldsymbol{\theta}}(\mathbf{x}), y)\right]$$
$$= R_i(f_{\boldsymbol{\theta}}) - \overline{R}(f_{\boldsymbol{\theta}}) - \epsilon.$$

Then, we can re-write ($P$-FL) as

$$\begin{aligned}
\mathsf{P}^{\star} = \min_{\theta \in \Theta} \quad & \mathbb{E}_{\overline{\mathcal{D}}}\left[\ell\big(f_{\theta}(\mathbf{x}), y\big)\right] \\
\text{subject to} \quad & \mathbb{E}_{\overline{\mathcal{D}}}\left[\ell_i\big(f_{\theta}(\mathbf{x}), y\big)\right] \leq 0, \quad i = 1, \ldots, m.
\end{aligned}$$
(PC-FL$'$)

which is in the form of (P).

As motivated in section 5.1, we then propose to solve the resilient version of ($P$-FL):

$$\begin{aligned}
P^{\star} = \min_{\boldsymbol{\theta} \in \Theta} \quad & R(f_{\boldsymbol{\theta}}) + h(\mathbf{u}) \\
\text{s. to} \quad & \mathbb{E}_{(\mathbf{x}, y) \sim \mathfrak{D}_i}\left[\ell(f_{\boldsymbol{\theta}}(\mathbf{x}), y) - R(f_{\boldsymbol{\theta}}) - \epsilon\right] \leq \mathbf{u}_i, \\
& i = 1, \ldots, C.
\end{aligned}$$
(PR-FL)

In the next section we derive the primal dual algorithm that enables us to tackle (PR-FL) in a distributed, privacy preserving manner.

## F.2 Resilient FL Algorithm

In this section we show that the resilient problem (PR-FL) can be tackled in a distributed and privacy preserving manner, as long as $h$ is additively separable in in each component. Explicitly, we assume that

$$h(\mathbf{u}) = \sum_i h_i(\mathbf{u}_i)$$

which holds for the quadratic cost used throughout the experimental section. We thus derive a distributed version of the resilient primal-dual algorithm presented in Section 4 (Algorithm 1).

We begin by writing the empirical Lagrangian associated to problem (PR-FL):

$$\hat{\mathcal{L}}(\theta, \boldsymbol{\lambda}, \mathbf{u}) = h(\mathbf{u}) + \frac{1}{C}\sum_{i=1}^{C} \frac{1}{N_i}\sum_{n_i=1}^{N_i}\left[\ell(f_{\boldsymbol{\theta}}(\mathbf{x}_{n_i}), y_{n_i})\right]$$
(57)

$$+ \lambda_i \left[\frac{1}{N_i}\sum_{n_i=1}^{N_i}\left[\ell(f_{\boldsymbol{\theta}}(\mathbf{x}_{n_i}), y_{n_i})\right] - \mathbf{u}_i\right]$$
(58)

$$= \sum_{i=1}^{C} \frac{1}{C}\left(1 + \lambda_i - \bar{\lambda}\right)\frac{1}{N_i}\sum_{n_i=1}^{N_i}\left[\ell(f_{\boldsymbol{\theta}}(\mathbf{x}_{n_i}), y_{n_i})\right] - \lambda_i \mathbf{u}_i + h_i(\mathbf{u}_i),$$
(59)

with $\bar{\lambda} = \frac{1}{C} \sum_{i=1}^{C} \lambda_i$.

Note that in (59) we have used the assumption that the perturbation cost is additively separable. This enables us to update the perturbation for each client *locally*, and thus the resilient formulation does not incurr in any additional communication costs (with respect to the constrained problem in [18]). That is, clients do not need to communicate $\mathbf{u}_i$ during optimisation. The server need only compute and communicate to all clients the average dual variable $\bar{\lambda}$ and the average loss $R(f_{\boldsymbol{\theta}})$. Therefore, Homomorphic Encryption techniques can be leveraged to compute these averages without revealing the values of individual dual variables $\lambda_i$ and local losses to the server. In the same manner, Homomorphic Encryption could be used to aggregate the relaxation cost $h(\mathbf{u}) = \sum_{i=1}^{X} h_i(\mathbf{u}_i)$ at the server without loss of privacy.

$\theta$ **update**. For a fixed $\boldsymbol{\lambda}$ and $\mathbf{u}$, the minimization of $\mathcal{L}$ with respect to $\theta$ is equivalent to a re-weighted version of the standard unconstrained FL objective:

$$\min_{\theta \in \Theta} \hat{\mathcal{L}}(\theta, \boldsymbol{\lambda}, \mathbf{u}) \iff \min_{\theta \in \Theta} \frac{1}{C} \sum_{i=1}^{C} \left(1 + \lambda_i - \bar{\lambda}\right) \frac{1}{N_i} \sum_{n_i=1}^{N_i} [\ell(f_{\boldsymbol{\theta}}(\mathbf{x}_{n_i}), y_{n_i})].$$

Therefore, $\theta$ can be updated using any FL solver. We thus refer to the update on $\theta$ as a subroutine ClientUpdate $\left(\{w_i\}_{i=1}^{C}, \theta^t\right) \to \theta^{t+1}$, which can be implemented using, for example FedAVG [56] or FedPD [57].

**u update**. For a fixed $\boldsymbol{\lambda}$ and $\theta$, the minimization of $\hat{\mathcal{L}}$ with respect to $\mathbf{u}$ is can be carried independently by each client, since the objective is additively separable:

$$\min_{\mathbf{u} \in \mathcal{U}} \hat{\mathcal{L}}(\theta, \boldsymbol{\lambda}, \mathbf{u}) \iff \min_{\mathbf{u} \in \mathcal{U}} \sum_{i=1}^{C} \lambda_i \mathbf{u}_i + h_i(\mathbf{u}_i)$$

We can thus employ gradient descent (or any other optimization algorithm) locally at each client.

$\boldsymbol{\lambda}$ **update**. For a fixed perturbation $\mathbf{u}$ and model $\theta$, we perform dual update on $\boldsymbol{\lambda}$ by taking dual ascent steps of the following equivalent objective,

$$\max_{\boldsymbol{\lambda} \in \mathbb{R}_{+}^{N}} \hat{\mathcal{L}}(\theta, \boldsymbol{\lambda}, \mathbf{u}) \iff \max_{\boldsymbol{\lambda} \in \mathbb{R}_{+}^{N}} \frac{1}{C} \sum_{i=1}^{C} \lambda_i \left( \frac{1}{N_i} \sum_{n_i=1}^{N_i} [\ell(f_{\boldsymbol{\theta}}(\mathbf{x}_{n_i}), y_{n_i})] - R(f(\theta)) - \mathbf{u}_i \right).$$

If server then computes and communicates the average of the client's losses $R(f(\theta))$, the updates to each multiplier $\boldsymbol{\lambda}_i$ can be made locally by each client as in 56.

As stated before, these update steps can be applied in alternated manner to find a solution of the empirical dual problem, as as described in Algorithm 2.

---

**Algorithm 2** Resilient Federated Learning

---

Initialize $\theta(0)$, $\boldsymbol{\lambda}(0)$, $\mathbf{u}(0)$, and $0 < \eta \ll 1$.
**for** $t = 1 \dots T$
    Compute weights: for all$i \in [N]$, $w_i = 1 + \lambda_i - \bar{\lambda}$, with $\bar{\lambda} = \frac{1}{N} \sum_{i=1}^{N} \lambda_i$;
    Theta Update: $\theta^t \leftarrow$ ClientUpdate $\left(\{w_i\}_{i=1}^{N}, \theta^{t-1}\right)$;
    Perturbation Update: $\mathbf{u}_i(t) = [\mathbf{u}_i(t-1) - \eta_{\mathbf{u}} d\mathbf{u}_i(t-1)]_{+}$
    Dual Update: $\boldsymbol{\lambda}_i(t) = \left[\boldsymbol{\lambda}_i(t-1) + \eta_{\boldsymbol{\lambda}} \left[\frac{1}{N_i} \sum_{j=1}^{N_i} \ell(f_{\boldsymbol{\theta}}(\mathbf{x}_j), y_j) - \mathbf{u}_i(t-1)\right]\right]_{+}$
**end**

---

### F.3 Experimental Setup

As in [18] we create hetereogeneity between clients through class imbalance. Unless stated otherwise, we use three minority classes (Class Labels $\{0, 2, 4\}$), and simulate the phenomenon of class imbalance by keeping only $\rho = 10\%$ of the data data belonging to the minority classes in the training set.

We follow the implementation from [58], in which a Dirichlet prior (over the ten classes) is sampled independently for each client. One client at a time, we sample without replacement according to each client's prior. Once a class runs out of samples, the subsequent clients do not own samples of that class.

We also create a test set for each client with the same class balance as the train set. Test sets are then sampled without replacement for each client (but with replacement among different clients, i.e., test sets overlap accross clients).

We use the same small neural network architecture as in [18, 58]. Namely, a CNN model consisting of 2 convolutional layers with $64 5 \times 5$ filters followed by 2 fully connected layers with 384 and 192 neurons. For both the constrained and resilient formulations, we use FedAVG [56] to update $\theta$, using 5 communication rounds for each primal update, as in [18].

In all experiments we use 100 clients, and all clients participate from each communication round. We run 500 communication rounds in total. We set $\epsilon = 0.1$, dual learning rate $\eta_\lambda = 0.1$, local learning rate $\eta_\theta = 5 \times 10^{-2}$ and use no data augmentation in all experiments. In the resilient learning formulation, unless stated otherwise, we use a quadratic penalty on the perturbation $h(u) = \|u\|_2^2$ and a perturbation learning rate $\eta_\mathbf{u} = 0.1$.

All of the plots in section 5 correspond to CIFAR10 [59]. Results for FashionMNIST [49] can be found on F.4.4.

## F.4   Additional Results

### F.4.1   Performance Relaxation trade off.

As we have already shown, through the choice of the relaxation cost function, we can control the trade-off between relaxing requirements and performance. Figure 4 shows that changing the ablation on the coefficient $\alpha$ in the quadratic relaxation cost $h(\mathbf{u}) = \alpha \|\mathbf{u}\|_2^2$ not only results in larger relaxations, but that these larger relaxations effectively lead to clients with higher losses.

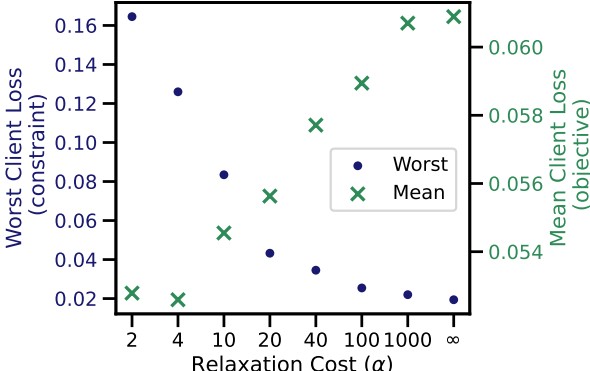

Figure 4: Train loss for the worst client and averaged over all clients for different values of the perturbation cost coefficient $\alpha$.

### F.4.2   Sensitivity to Problem Specification

The resilient approach is less sensitive to the specification of the constraints. Dual variables indicate the sensitivity of the objective with respect to constraint perturbations. As shown by Figure 5 the resilient approach yields smaller dual variables, irespectively of the tolerance $\epsilon$ in the constraint specification. In this context, setting the constraint levels a priori requires knowledge about the heterogeneity of the client local distributions, which in our setup depends on the imbalance of the dataset. As shown by figure 6 the resilient approach yields smaller dual variables, irrespectively of the tolerance irrespectively of the percentage of samples of minority classes that are kept, whereas the dual variables grow for more imbalanced and thus harder to satisfy scenarios.

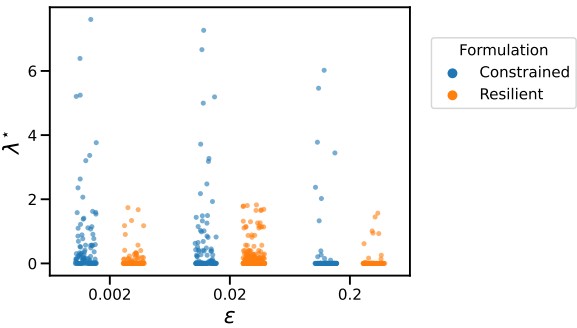

Figure 5: Dual variables after training with respect to constraint specification $\epsilon$.

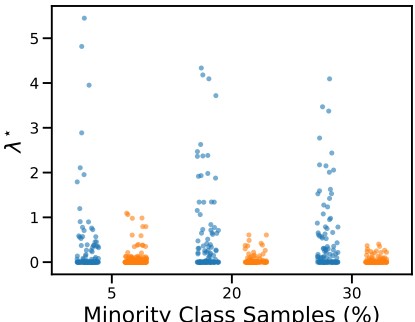

Figure 6: Dual variables at the end of training with respect to the percentage of minority samples $\rho$ kept in the dataset. Smaller $\rho$ means more imbalance, and thus results in harder to satisfy requirements.

### F.4.3 Constraint violation across setups.

Relaxing stringent requirements not only makes the empirical problem easier to solve, but it can also lead to a better empirical approximation of the underlying statistical problem. That is, overly stringent requirements can result not only in large dual variables, but can also harm generalization. Figure 7 shows that more stringent constraint specifications can lead to poor constraint satisfaction in the test set. This improvement in constraint satisfaction is associated with smaller generalization gaps for the constraints were observed for the resilient approach, as shown in Figure 8.

### F.4.4 Test Accuracy.

In this section we aim to give quantitative performance metrics. We include a constrained baseline [18] and another method for federated learning under class imbalance [60] for comparison.

We first compare approaches in terms of the objective, by averaging the test accuracy over all clients. As shown in Table 3, the average accuracy of the resilient approach is overall similar to the constrained approach and the baseline [60].

In order to asses how the distribution o performance among clients varies we also report spread metrics for the test accuracy across clients. As shown in Table 3, Resilient learning has (generally) less spread in the interquartile range and higher maximum spread than its constrained counterpart.

### F.4.5 Comparing client performances

We sort clients according to their test accuracy $\text{Acc}_{[1]} \geq \text{Acc}_{[2]} \geq \ldots \geq \text{Acc}_{[c]}$. We plot the accuracy of equally ranked clients for the constrained baseline [18] ($\text{Acc}_{[c]}^{\text{res}} \geq \text{Acc}_{[c]}^{\text{const}}$). As shown in Figures 9 and 10, in all setups the majority of clients achieve a small increase in performance, while a small fraction experiences a larger decrease. For higher imbalance ratios, this is more pronounced.

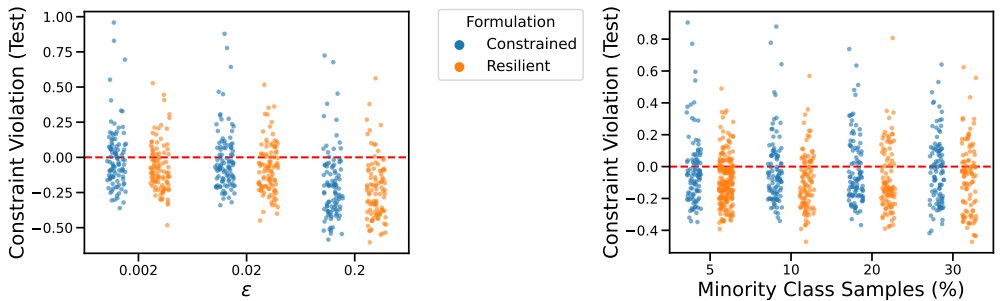

Figure 7: Constraint violation (evaluated no the test set) for Resilient and constrained learning using different (left) constraint specifications $\epsilon$, and (right) fraction of minority classes.

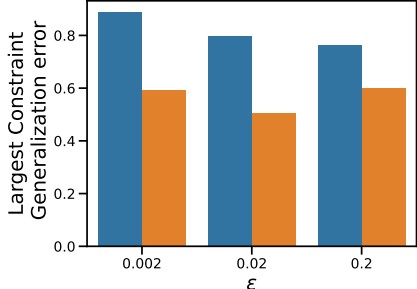

Figure 8: Largest generalization error across clients for different constraint specifications $\epsilon$.

### F.5 Ablation on perturbation cost function $h$.

In all experiments, we have used a quadratic perturbation cost function $h = \alpha\|u\|_2^2$ In order to assess the impact of the choice of cost the function, we run our algorithm using as a cost $\|u\|_\beta$ with $\beta = 1, 2, 4, \infty$, for fashion MNIST in both the federated and invariant setting.

As shown in Table 4, both the mean and spread statistics are similar for $\beta = 1, 2, 4$, i.e. our approach is not overly sensitive to the choice of cost function in this experimental setup. However, $\beta = \infty$ does show a substantially different behaviour. Penalizing the only the largest perturbation results in a smaller range while attaining a higher IQR and lower mean accuracy. That is, it reduces the worst client test accuracy, but the performance of most clients deteriorates, both in terms of its average and spread. The infinity norm is related to Rawlsian (i.e. minimax) formulations which have been proposed in the context of fairness see e.g. [14].

## G Ablation on dual and resilient learning rates

We also perform an ablation on $\eta_u, \eta_\lambda$, the perturbation and dual learning rates, over a small grid of 12 values. In this particular setup, we find that the performance of the algorithm is not overly sensitive to this choice. We also observe that that the rates that were used in the paper ($\eta_u = 0.1, \eta_\lambda = 2$) are not optimal in this case, and thus further improvements in performance could be obtained through

| Dataset | Imb. Ratio | Mean | | | IQR | | | Range | | |
|---|---|---|---|---|---|---|---|---|---|---|
| | | [60] | [18] | Ours | [60] | [18] | Ours | [60] | [18] | Ours |
| F-MNIST | 10 | 92.5 | 92.6 | 93.4 | 2.2 | 3.5 | 2.4 | 64.1 | 28.6 | 50.6 |
| | 20 | 94 | 93.8 | 94.4 | 2.2 | 2.7 | 1.7 | 50.2 | 28.6 | 45.8 |
| CIFAR10 | 10 | 81.3 | 81.5 | 81.5 | 8.1 | 8.7 | 8.7 | 49.4 | 35.8 | 46.4 |
| | 20 | 83.4 | 82.4 | 82.6 | 8.6 | 8.4 | 7.9 | 44.5 | 32 | 41.1 |

Table 3: Client accuracy spread metrics for different setups. IQR denotes interquantile range and range denotes the maximum minus the minimum accuracy, both computed across 100 clients.

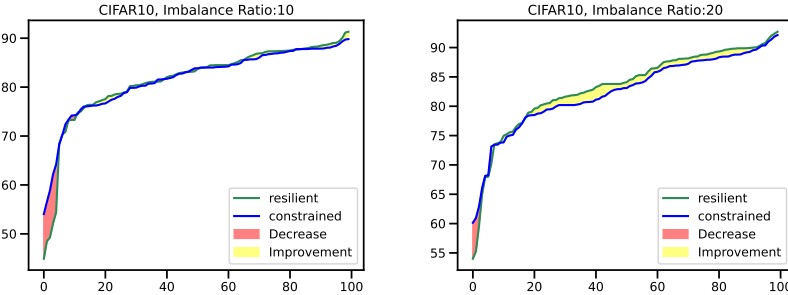

Figure 9: Test Accuracy for equally ranked clients for the resilient and constrained baseline [18] in CIFAR10.

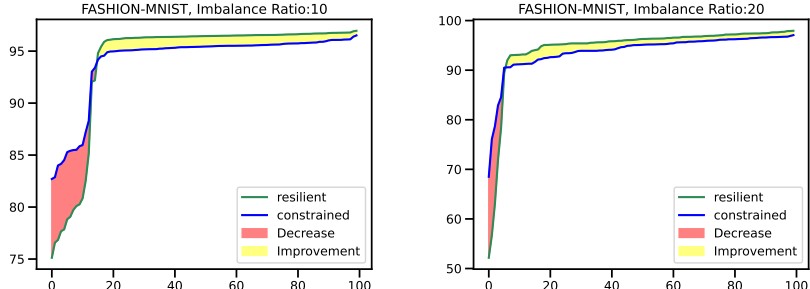

Figure 10: Test Accuracy for equally ranked clients for the resilient and constrained baseline [18] in CIFAR10.

more extensive hyperparameter tuning. However, the main aim of our numerical experiments is to validate empirically the properties of our approach.

# H    Invariance Constrained Learning Experiments

## H.1    Experimental setup.

We showcase our approach on datasets with artificial invariances, following the setup of [23]. Explicitly, we generate the synthetic datasets, by applying either rotations, translations or scalings, to each sample in the MNIST [48] and FashionMNIST [49] datasets. The transformations are sampled from uniform distributions over the ranges detailed in Table 6. We use the same MLP and CNN architectures and hyperparameters as [23], except that we use only 6 augmentations per sample instead of 31 during training.

For each transformation set (rotations, translations and scalings), we constraint the expected loss over samples augmented with the transformations sampled from uniform distributions over the ranges detailed in Table 7. Note that there is a mismatch between the distribution used to generate the

| $\beta$ | Mean Acc. | IQR | Max Range |
|---|---|---|---|
| 1.0 | 93.3 | 2.5 | 49.5 |
| 2.0 | 93.4 | 2.4 | 50.6 |
| 4.0 | 93.4 | 2.6 | 45.6 |
| $\infty$ | 92.7 | 3.8 | 28.1 |

Table 4: Cost function ablation for the heterogeneous federated learning in fashion-MNIST using 100 clients, 3 minority classes, an imbalance ratio of $\rho = 10$ and dirichlet allocation with parameter $d = 0.3$. We report the mean, interquartile range and range (maximum value minus minimum value) test accuracy across clients.

| $\eta_u \backslash \eta_\lambda$ | 0.1 | 0.5 | 1 | 2 |
|---|---|---|---|---|
| 0.1 | 81.4 | 81.6 | 81.8 | 81.8 |
| 0.5 | 81.4 | 80.9 | 81.4 | 81.1 |
| 1 | 81.6 | 81.6 | 81.6 | 81.7 |

Table 5: Dual and resilient learning rate ablation in Heterogenous federated learning setting. We report mean Test Accuracy for CIFAR100 using 100 clients, 3 minority classes, an imbalance ratio of $\rho = 10$ and dirichlet allocation with parameter $d = 0.3$.

data and that used in the constraints. That is, except for the fully rotated dataset, the constraints are larger than the true transformation range used to construct the synthetic dataset (Table 6).The purpose of this experiments is to showcase that the resilient approach can relax constraints associated to transformation sets that do not describe symmetries or invariances of the data and can thus hinder performance. We use the same transformation sets and constraint specification ($\epsilon$) for all synthetic datasets.

| Synthetic invariance | Parameter | Distribution |
|---|---|---|
| Full Rotation | Angle in radians. | $\mathcal{U}\left[-\frac{\pi}{2}, \frac{\pi}{2}\right]$ |
| Partial Rotation | Angle in radians. | $\mathcal{U}[-\pi, \pi]$ |
| Translation | Translation in pixels. | $\mathcal{U}[-8, 8]^2$ |
| Scale | Exponential Scaling factor. | $\mathcal{U}[-log(2), log(2)]$ |

Table 6: Sampling parameters for transformations used to obtain synthetically invariant datasets, from [23]

.

| Constraint Set | Parameter | Range |
|---|---|---|
| Rotations | Angle in radians. | $[-\pi, \pi]$ |
| Translation | Translation in pixels. | $[-16, 16]^2$ |
| Scale | Exponential Scaling factor. | $[-1.5, 1.5]$ |

Table 7: Transformation sets $\mathcal{G}_i$ used as invariance constraints. All sets are used simultanously, with the same constraint level ($\epsilon_i$) for all datasets (0.1).

## H.2 Results

### H.2.1 Ablation on perturbation cost.

As in the case of federated learning (presented in section 4) perform an ablation on the cost function $h$ using $\|u\|_\beta$ with $\beta = 1, 2, 4, \infty$.

As shown in table 8, $\beta = 1$ showed slightly better performance in terms of average test accuracy across all invariant settings. As discussed in Section 3.3, $\beta = 1$ recovers penalty based methods, i.e. it is equivalent to setting a fixed penalization coefficient. Nonetheless, the dynamics of our algorithm are different and can thus lead to different solutions.

| $\beta$ | Partially Rotated | Translated | Scaled |
|---|---|---|---|
| 1.0 | $86.08 \pm 0.38$ | $86.85 \pm 0.20$ | $85.02 \pm 0.46$ |
| 2.0 | $85.37 \pm 0.17$ | $86.65 \pm 0.25$ | $84.92 \pm 0.39$ |
| 4.0 | $85.23 \pm 0.20$ | $86.64 \pm 0.11$ | $84.65 \pm 0.68$ |
| $\infty$ | $82.94 \pm 0.17$ | $85.47 \pm 0.19$ | $83.35 \pm 0.49$ |

Table 8: Cost function ablation for invariant fashion-MNIST datasets. We compute the mean and standard deviation in test accuracy across three independent runs.

### H.2.2 Performance on F-MNIST

The resilient approach is able to handle the misspecification of invariance requirements, outperforming in terms of test accuracy both the constrained and unconstrained approaches. In addition, though it was not designed for that purpose, our approach shows similar performance to the invariance learning method Augerino [22].

| Dataset | Method | Rotated (180) | Rotated (90) | Translated | Scaled | Original |
|---------|--------|---------------|--------------|------------|--------|----------|
| F-MNIST | Augerino | **85.28 ± 0.54** | 81.48 ± 0.49 | 81.13 ± 0.77 | 83.17 ± 0.46 | 90.09 ± 0.20 |
| | Unconstrained | 77.94 ± 0.06 | 81.57 ± 0.36 | 79.23 ± 0.17 | 82.99 ± 0.18 | 90.20 ± 0.23 |
| | Constrained | 84.96 ± 0.12 | 85.66 ± 0.32 | 83.61 ± 0.10 | 86.49 ± 0.09 | 91.02 ± 0.02 |
| | Resilient | **85.57 ± 0.26** | **86.48 ± 0.15** | **85.06 ± 0.23** | **87.26 ± 0.14** | **91.55 ± 0.31** |

Table 9: Classification accuracy for synthetically invariant F-MNIST. We use the same invariance constraint level $\epsilon_i = 0.1$ for all datasets and transformations. We include the invariance learning method Augerino [22] as a baseline.

