# OpenReview forum: "Resilient Constrained Learning"
_NeurIPS.cc/2023/Conference — NeurIPS 2023 poster_

### Official Review · Reviewer_4YtT · 2023-06-29

**Soundness:** 3 good
**Presentation:** 3 good
**Contribution:** 3 good
**Rating:** 6
**Confidence:** 3

**Summary:**

Reasonable requirement specification in constrained learning has long been hindered by the presence of compromises and limited prior knowledge about the data. As a treatment, this paper proposes resilient constrained learning that adapts the requirements while simultaneously solving the learning task. Specifically, it balances the performance gains obtained from the constraint relaxation against a user-defined relaxation cost function. The paper provides theoretical provements for the balance (e.g. approximation and generalization guarantees) along with a practical algorithm. The algorithm is validated in invariant learning and federated learning experiments.

**Strengths:**

1. The paper is well motivated to solve the constraint-performance trade-off problem, with the algorithm derived from detailed assumptions and theorems.
2. Overall, I think the paper is well-written and easy-to-follow.
3. The final algorithm is simple yet effective, allowing for a straightforward and flexible adjustment of the trade-off through a relaxation cost parameter $\alpha$.

**Weaknesses:**

1. My primary concern regarding the acceptance of this paper lies in its simplistic experimentation. From my view, the federated learning problem is more like a toy example.
2. The paper lacks in-depth study on various aspects of the algorithm's performance.

**Questions:**

1. Have you try cost functions other than $\alpha||u||^2$, and how does different cost functions impact the algorithm's performance and efficiency?
2. How does the learning rates $\eta, \eta_u,\eta_\lambda$ influence the algorithm's performance? Do they require meticulous joint tuning, or are they effective within certain ranges of individual values?

**Limitations:**

1. First, the federated and invariance learning problem are manually synthesized for validating the effectiveness of the proposed method, which I regard not convincing enough without results on large-scale real-world datasets.
2. The federated learning experiment lacks quantitative metrics and diverse baselines other than single constrained learning.
3. The empirical efficiency has not been studied in the paper.
4. The paper lacks a detailed limitation discussion section.

---

> ### Author Rebuttal · Authors · 2023-08-09
>
> We thank the reviewer for their valuable feedback. In what follows, we address the points raised by the reviewer.
>
> We respectfully disagree with the claim that the experimentation conducted is *(Weaknesses 1) simplistic*. Both the federated and invariant learning are benchmark setups used in recent literature [R4, R5]. We also disagree on the claim that we did not conduct several experiments that assess *(Weaknesses 2) various aspects of the algorithm's performance*. In both setups, we analyze several properties of our method, that validate empirically the motivating theory, for example:
> - *Figure 2 (left) and 8*: more stringent requirements are relaxed more
> - *Figure  2 (right), 7,  8, Table 1 and 4*: these relaxations lead to an improvement in performance.
>
>
> In what follows, we provide a detailed discussion of new experimental results addressing the reviewers concerns.
>
> In the federated setup, we adopt as benchmarks Ratio-Loss [R2] and CLIMB [R3]. The latter is a constrained learning approach which we adopt with the modification that constrains cab be relaxed as per the definition of resilient equilibrium. In this setup, average accuracy is not really the pertinent metric. Observe that the *average* accuracy (Response Table 1) is  overall similar to the constrained approach, and consistently higher than the baseline [R3].
>
> However, the distribution o performance among clients varies. We actually chose this experiment as one in which the differences between resilient learning and standard constrained learning are apparent. Resilient learning has (generally) less spread in the interquartile range and higher maximum spread (Response Table 2) than the constrained approach.
>
> This is precisely what the method is designed to do. Sacrifice the performance of outliers to benefit the performance of the majority of the agents. In order to showcase this, we order clients by their accuracy. We then compute the fraction of clients in the resilient method that outperform equally ranked clients for baseline methods (Response Table 3).
>
> ## New Ablations
>
> ### Choice of cost function
>  We run $||u||_\beta$ with $\beta =1, 2, 4$, and infinity, for fashion MNIST in both the federated and invariant setting. As discussed in Section 3.3, $\beta = 1$ recovers penalty based methods.
>  **Federated Learning**
> | $\beta$    | Mean Acc | IQR   | Max Range |
> |------------|----------|-------|-----------|
> | 1.0        | $93.3$   | $2.5$ | $49.5$    |
> | 2.0        | $93.4$   | $2.4$ | $50.6$    |
> | 4.0        | $93.4$   | $2.6$ | $45.6$    |
> | $\infty$   | $92.7$   | $3.8$ | $28.1$    |
>
> **Response Table 4**:*Cost function ablation for the heterogeneus federated learning in fashion-MNIST using 100 clients, 3 minority classes, an imbalance ratio of $\rho=10$ and dirichlet allocation with parameter $d=0.3$. We report the mean, interquartile range and range (maximum value minus minimum value) test accuracy across clients.*
>
> **Invariance**
>
>  | $\beta$  | Partially Rotated | Translated      | Scaled          |
> |----------|-------------------|-----------------|-----------------|
> | 1.0      | $86.08\pm 0.38$   | $86.85\pm 0.20$ | $85.02\pm 0.46$ |
> | 2.0      | $85.37\pm 0.17$   | $86.65\pm 0.25$ | $84.92\pm 0.39$ |
> | 4.0      | $85.23\pm 0.20$   | $86.64\pm 0.11$ | $84.65\pm 0.68$ |
> | $\infty$ | $82.94\pm 0.17$   | $85.47\pm 0.19$ | $83.35\pm 0.49$ |
>
> **Response Table 5**:*Cost function ablation for invariant fashion-MNIST datasets. We compute the mean and standard deviation in test accuracy across three independent runs.*
>
> ### Learning Rates
> We run an ablation on $\eta_u, \eta_\lambda$, the perturbation and dual learning rates over a small grid of 12 values, and find that, in this range, the performance of the algorithm is not overly sensitive to this choice. We also observe that  that the rates that were used in the paper ($\eta_u = 0.1, \eta_\lambda = 2$) are not optimal in this setup, and thus further improvements in performance could be obtained through more extensive hyperparameter tuning, but this was not the focus of our experiments.
>
> | $\eta_u$ \ $\eta_\lambda$ | 0.1  | 0.5  | 1    | 2    |
> |-----------------------|------|------|------|------|
> | 0.1                   | 81.4 | 81.6 | 81.8  | 81.8 |
> | 0.5                   | 81.4 | 80.9 | 81.4 | 81.1 |
> | 1                     | 81.6 | 81.6 | 81.6 | 81.7 |
>
> **Response Table 6**:*Dual and resilient learning rate ablation in Heterogenous federated learning setting. We report mean Test Accuracy for CIFAR100 using 100 clients, 3 minority classes, an imbalance ratio of $\rho=10$ and dirichlet allocation with parameter $d=0.3$.*
>
>  *References*
>
> [R1] Zhu, Hangyu, et al. "Federated learning on non-IID data: A survey." Neurocomputing 465 (2021): 371-390.
>  [R2] Wang, Lixu, et al. "Addressing class imbalance in federated learning." Proceedings of the AAAI Conference on Artificial Intelligence. Vol. 35. No. 11. 2021.
>  [R3] Shen, Zebang, et al. "An agnostic approach to federated learning with class imbalance." International Conference on Learning Representations. 2021.
>  [R4] Durmus, Alp Emre, et al. "Federated Learning Based on Dynamic Regularization." International Conference on Learning Representations. 2021.
>  [R4] Durmus, Alp Emre, et al. "Federated Learning Based on Dynamic Regularization." International Conference on Learning Representations. 2021.
>  [R5] Immer, Alexander, et al. "Invariance learning in deep neural networks with differentiable Laplace approximations." Advances in Neural Information Processing Systems 35 (2022): 12449-12463.

---

> > ### Comment · Reviewer_4YtT · 2023-08-10
> >
> > Thank you for the feedback. I'm pleased with the new experimental results and your explanation about my confusions. I will raise my score to 6.

---

### Official Review · Reviewer_69dA · 2023-07-08

**Soundness:** 3 good
**Presentation:** 2 fair
**Contribution:** 3 good
**Rating:** 5
**Confidence:** 3

**Summary:**

This paper proposes an approach to solve the constrained learning problem (where the considered function is convex), named resilient constrained learning. The main idea is to relax the learning constraints according to how much they affect the considered task. It can be viewed as a generalization of the standard constrained learning and studies the tradeoff between the gain from constraint relaxations and the cost from the relaxations. The main theoretical result shows that the gap between the relaxed solution and the optimal solution of the original problem is bounded by a function of the relaxation amount. Numerical experiments verified the basic properties of the proposed approach.

**Strengths:**

The idea of relaxing the constraints and balance between relaxation gain and cost is interesting. When $u$>0, it can be seen as an interpolation between unconstrained learning and constrained learning. The main theorem states a bound for the gap between the relaxed solution and the original optimal solution and the derivation process seems solid (although I did not check every step carefully). It can have meaningful real life applications since in many real cases, the constraints are considered as compromisable depending on the gains of relaxations.

**Weaknesses:**

The writing is a little hard to follow and some important parts are put in the appendix, like the algorithm 2 mentioned below theorem 1.

One major concern is that the experimental results are quite weak even considering that this is a theoretical paper. The experiments are only sanity checks to make sure the proposed method can indeed solve some constrained learning problems, but did not demonstrate how well the solution is. There is no comparison with existing algorithms for solving constrained learning problems, which hinders the contribution of the proposed method to a large extent.

**Questions:**

It would be much better if there were some comparison with existing constrained learning algorithms, and discuss the cases where $u=0$ and $u>0$.

**Limitations:**

Did not see discussions of limitations.

---

> ### Author Rebuttal · Authors · 2023-08-09
>
> We thank the reviewer for their valuable feedback. In what follows, we address the main points raised by the reviewer.
>
> *some important parts are put in the appendix, like the algorithm 2* The algorithm referenced below theorem 1 should be Algorithm 1, which is indeed included in the main text. Algorithm 2 in the appendix pertains the federated setup only (i.e., it is more specific).
>
> *There is no comparison with existing algorithms for solving constrained learning problems*.
> We do compare with existing primal-dual algorithms for solving constrained learning problems. This algorithm has been used in recent works in both the imbalanced federated learning [R3] and invariant learning [R5] setup. All of our experimental  comparisons include constrained learning. In addition, we have added new experiments in order to give a more detailed quantitative comparison between methods.
>
> The main concern is that we *did not demonstrate how well the solution is*. In that sense, we reported performance on Tables 1 (Imbalanced-FL) and 4(Invariant Learning), showing that our method performs well compared to baselines. Below, we add a more in depth discussion of existing and new results.
>
> ### New results in rebuttal
>
> We point out that federated learning with class imbalance, also known as heterogeneous federated learning, is a problem motivated by practical considerations which has received substantial attention; see e.g [R1] and references therein. We adopt it as an example in this paper as a prototypical situation in which accommodating some clients -- i.e., constraints -- can be much more difficult than accomodating most clients.
>
> We adopt as benchmarks Ratio-Loss [R2] and CLIMB [R3]. The latter is a constrained learning approach which we adopt with the modification that constrains cab be relaxed as per the definition of resilient equilibrium.
>
>  Also, we did not tune any hyperparameters and used the same hyperparameters as in [R3], since the objective was to provide a fair comparison. In order to see how the hyperparameters that are exclusive to our method, namely $\eta_u$ and $h(u)$, we conducted ablations detailed below.
>
> We also thank the reviewer for their suggestion to include more quantitative metrics. In this setup, average accuracy is not really the pertinent metric. Observe that the *average* accuracy (Response Table 1) is  overall similar to the constrained approach, and consistently higher than the baseline [R3].
>
> However, the distribution o performance among clients varies. We actually chose this experiment as one in which the differences between resilient learning and standard constrained learning are apparent. Resilient learning has (generally) less spread in the interquartile range and higher maximum spread (Response Table 2) than the constrained approach.
>
> This is precisely what the method is designed to do. Sacrifice the performance of outliers to benefit the performance of the majority of the agents. In order to showcase this, we order clients by their accuracy. We then compute the fraction of clients in the resilient method that outperform equally ranked clients for baseline methods (Response Table 3).
>
> *References*
>
> [R1] Zhu, Hangyu, et al. "Federated learning on non-IID data: A survey." Neurocomputing 465 (2021): 371-390.
>  [R2] Wang, Lixu, et al. "Addressing class imbalance in federated learning." Proceedings of the AAAI Conference on Artificial Intelligence. Vol. 35. No. 11. 2021.
>  [R3] Shen, Zebang, et al. "An agnostic approach to federated learning with class imbalance." International Conference on Learning Representations. 2021.
>  [R4] Durmus, Alp Emre, et al. "Federated Learning Based on Dynamic Regularization." International Conference on Learning Representations. 2021.
>
> ## Comparisons to constrained learning and discussion.
>
> In addition to the quantitative metrics presented above, the results included both in the experimental section and appendix mainly aim to highlight the differences between our approach and existing constrained learning approaches. In doing so, we discuss the inherent trade-offs or limitations of out approach. Here we provide a brief summary and expand on how these address the fundamental aspects of our method.
>
> **Constraint Relaxation and Relative difficulty**:
> This illustrates how our method adapts $u$ depending on how difficult it is to satisfy the constraint, whereas the constrained approach enforces the constraint ($u=0$) regardless of the price to pay in performance. Since the impact on performance depends on the data distribution, the most imbalanced clients (Figure 2 in main text) are relaxed more. This shows empirically that our method behaves as intended *in a learning setup*. The same happens for invariances that are not present in the dataset (Figure 8 in Appendix).
>
> **Controlling the performance  vs. relaxation trade-off**:
> These experiments (Figure 2 in main text, Figure 4 in the Appendix) illustrate the inherent trade-off between imposing constraints and having better performance in terms of statistical risk in the objective.
>
> **Sensitivity to Problem Specification**:
> These experiments (Figure 3 left in main text, Figure 6 in the appendix) highlight that our method effectively eases the challenge of specifying constraint levels in both practical setups.
>
> **Constraint violation and Generalization**:
>  Based on our theoretical approximation bounds, resilient learning should have better generalization. These experiments (Figure 3 right in main text, Figure 6 in the Appendix) show that this holds in practice.

---

> > ### Comment · Reviewer_69dA · 2023-08-18
> >
> > Thank you for the detailed response. The comparison with existing constrained learning methods now look reasonable and I have increased my score.

---

### Official Review · Reviewer_iyNP · 2023-07-21

**Soundness:** 3 good
**Presentation:** 3 good
**Contribution:** 3 good
**Rating:** 5
**Confidence:** 3

**Summary:**

This paper introduces the concept of resilient constrained learning, which aims to find a compromise between reducing the objective loss and staying close to the original problem. This paper presents conditions for achieving a resilient equilibrium and provides equivalent formulations of resilient relaxation. This paper derives approximation and statistical guarantees for the algorithm used to find this equilibrium. It showcases its advantages in image classification tasks involving multiple potential invariances and federated learning under distribution shift. The paper's contributions include introducing a practical algorithm to compute the resilient equilibrium, determining conditions under which this equilibrium exists, and showcasing the advantages of resilient constrained learning in real-world applications.



**Strengths:**

+ In terms of originality, resilient constrained learning is introduced, a novel approach to balancing multiple requirements in machine learning tasks. The idea of adapting the requirements while simultaneously solving the learning task is innovative. It addresses the challenge of specifying requirements in the presence of compromises and limited prior knowledge about the data. The paper also presents conditions for achieving a resilient equilibrium and provides equivalent formulations of resilient relaxation.

+ The paper's significance lies in its potential to enable machine learning solutions that better satisfy real-world requirements beyond accuracy, such as fairness, robustness, or safety.



**Weaknesses:**



- One weakness is that the paper does not comprehensively compare existing approaches to constrained learning, such as penalty-based methods or Lagrangian duality-based methods. While the article mentions these approaches, it does not compare their strengths and weaknesses with the Resilient Constrained Learning approach. Such a comparison could help clarify the advantages and limitations of the Resilient Constrained Learning approach and provide insights into when it is most appropriate.

- Additionally, the paper could benefit from a more detailed discussion of the limitations and assumptions of the proposed method.

- From my humble understanding, the inequality in eq (1) should be P(v) >= P(u) + p\top (v - u_0).

- Grammar issues: L 39. be they penalty coefficients or constraint levels

**Questions:**

See Above

---

> ### Author Rebuttal · Authors · 2023-08-09
>
> We thank the reviewer for their valuable feedback. In what follows, we address the points raised by the reviewer.
>
> *... the inequality in eq (1) should be...* Thanks for the observation, it was a typo.
>
> *...the paper does not comprehensively compare existing approaches to constrained learning,...*
>
> We do compare with existing primal-dual algorithms for solving constrained learning problems. These algorithms have been used in recent works in both the imbalanced federated learning [R3] and invariant learning [R5] setup. All of our experimental  comparisons include constrained learning:
>
> -  Resilient solutions sacrifice stringent constraints to improve overall performance. For instance, the results shown in Figures 2(left) and 8 (in the supplementary materials) demonstrate that more stringent requirements are relaxed more and Figures 2 (right), 7 and 8 along with Tables 1 and 4 demonstrate that these relaxations lead to performance improvements as measured by the objective loss.
> - We have also included several numerical experiments to highlight properties of resilient constrained learning. For instance, Figures 3 (left), 5 and 6 show decreased sensitivity to problem specification and Figures 3 (right) and 7 show better empirical approximation of the underlying statistical problem. Both of these are properties that resilient cosntrained learning has by definition.
>
> In addition, we have included new results and comparisons to further highlight the differences between our approach and constrained learning, as detailed below.
>
> We adopt as benchmarks Ratio-Loss [R2] and CLIMB [R3]. The latter is a constrained learning approach which we adopt with the modification that constrains can be relaxed as per the definition of resilient equilibrium.
>
>  Also, we did not tune any hyperparameters and used the same hyperparameters as in [R3], since the objective was to provide a fair comparison. In order to see how the hyperparameters that are exclusive to our method, namely $\eta_u$ and $h(u)$, we conducted ablations detailed below.
>
> In this setup, average accuracy is not really the pertinent metric. Observe that the *average* accuracy (Response Table 1) is  overall similar to the constrained approach, and consistently higher than the baseline [R3].
> | Dataset | Imb. Ratio | Num Minority | Ratio-Loss[R2] | CLIMB[R3]  | OURS   |
> |---------|------------|--------------|------------|--------|--------|
> | F-MNIST | 10         | 3            | $92.5$     | $92.6$ | $93.4$ |
> | F-MNIST | 20         | 3            | $94.0$     | $93.8$ | $94.4$ |
> | CIFAR10 | 10         | 3            | $81.3$     | $81.5$ | $81.5$ |
> | CIFAR10 | 20         | 3            | $83.4$     | $82.4$ | $82.6$ |
>
> **Response Table 1**: Average  Accuracy for different setups. The imbalance ratio denotes the fraction of samples kept in minority classes. As in [R4] we use a dirichlet distribution to allocate samples among clients, with parameter $d=0.3$.
>
> However, the distribution o performance among clients varies. We actually chose this experiment as one in which the differences between resilient learning and standard constrained learning are apparent. Resilient learning has (generally) less spread in the interquartile range and higher maximum spread (Response Table 2) than the constrained approach.
>
> | Dataset | Imb. Ratio | Ratio Loss[R2]     | CLIMB[R3]          | OURS           |
> |---------|------------|----------------|----------------|----------------|
> | F-MNIST | 0.1        | $2.2$ $(64.1)$ | $3.5$ $(28.6)$ | $2.4$ $(50.6)$ |
> | F-MNIST | 0.05       | $2.2$ $(50.2)$ | $2.7$ $(28.6)$ | $1.7$ $(45.8)$ |
> | CIFAR10 | 0.1        | $8.1$ $(49.4)$ | $8.7$ $(35.8)$ | $8.7$ $(46.4)$ |
> | CIFAR10 | 0.05       | $8.6$ $(44.5)$ | $8.4$ $(32.0)$ | $7.9$ $(41.1)$ |
>
> **Response Table 2**: Client accuracy spread metrics for different setups. The first number denotes interquantile range and the number in parentheses denotes the maximum minus the minimum accuracy, both computed across 100 clients.
>
> This is precisely what the method is designed to do. Sacrifice the performance of outliers to benefit the performance of the majority of the agents. In order to showcase this, we order clients by their accuracy. We then compute the fraction of clients in the resilient method that outperform equally ranked clients for baseline methods (Response Table 3).
> | Dataset | Imb. Ratio | Improved (%) | Mean Improvement | Max Improvement | Mean Decrease | Max Decrease |
> |---------|------------|--------------|------------------|-----------------|---------------|--------------|
> | CIFAR10 | 10         | 77           | 0.4              | 1.5             | 0.5           | 10.0         |
> | CIFAR10 | 20         | 79           | 0.5              | 2.1             | 0.3           | 9.1          |
> | F-MNIST | 10         | 92           | 1.6              | 4.8             | 0.9           | 23.3         |
> | F-MNIST | 20         | 94           | 1.3              | 2.6             | 0.7           | 19.4         |
>
> **Response Table 3**: Changes in accuracy for equally ranked clients for the resilient method.
> As intended, performance improves for most clients, though at the cost of a decrease in performance for a few *outliers*.
>
> *References*
>
> [R1] "Federated learning on non-IID data: A survey." Neurocomputing (2021).
>  [R2]"Addressing class imbalance in federated learning." AAAI 2021.
>  [R3] "An agnostic approach to federated learning with class imbalance." ICLR 2021.
>  [R4] "Federated Learning Based on Dynamic Regularization." ICLR 2021.
> [R5] "Automatic data augmentation via invariance-constrained learning." ICML 2023.

---

> > ### Comment · Area_Chair_oE63 · 2023-08-20
> > **To Reviewer iyNP: Please respond to the author rebuttal**
> >
> > Dear Reviewer iyNP,
> >
> > The deadline for author discussion period is approaching soon. Please respond to the author's rebuttal and indicate whether your concerns have been addressed. Thank you!
> >
> > -AC

---

### Official Review · Reviewer_tedN · 2023-07-27

**Soundness:** 3 good
**Presentation:** 3 good
**Contribution:** 4 excellent
**Rating:** 7
**Confidence:** 3

**Summary:**

This paper proposes a novel resilient learning approach for constrained learning problem. In the presented approach, constraints are interpreted as nominal specification that can be relaxed to find a better compromise between objective and requirements. The first main contribution of this paper is to relax constraints according to the sensitivity of the objective to perturbations of the constraint. Then the next contribution is to design a practical resilient learning algorithm based on duality and perturbation theory.

**Strengths:**

1.	The assumption and properties of the convex function of relaxation and resilient equilibrium are discussed in detail in this paper, which provides a theoretical proof to the effectiveness of the presented approach.
2.	The authors also interpret why both traditional unconstrained and constrained learning can be seen as limiting cases of resilient learning.
3.	The paper is well written.


**Weaknesses:**

1.	Some basic concepts should be explained more clearly, such as the definition of nominal specification.
2.	The experiment of invariant learning is missed in Section 5 although in the abstract it is claimed to has been conducted.
3.	The authors do not explain which constrained approaches are involved in the contrast experiment.


**Questions:**

I cannot find the evaluation of resilient formulation of invariant learning. Where is it?

**Limitations:**

No limitations or potential negative societal impact are mentioned.

---

> ### Author Rebuttal · Authors · 2023-08-09
>
> We thank the reviewer for their feedback. In what follows, we address the main points raised by the author.
>
> *(Weakness 1) Some basic concepts should be explained more clearly, such as the definition of nominal specification.* Thanks, we will add the clarification that nominal specification means u = 0.
>
> *(Weakness 2) The experiment of invariant learning is missed in Section 5 although in the abstract it is claimed to has been conducted.*
>
> As stated at the start of Section 5 (line 272) experimental results for invariance learning are included in appendix G. We reproduce some of the results below for convenience.
>
> *The authors do not explain which constrained approaches are involved in the contrast experiment* We compare with an existing primal-dual alternating algorithm as previously used for both the imbalanced federated learning [1] and inveriant learning [2] setup. Note that this algorithm is a particular case of ours (i.e., when $h$ is the indicator function of the non-negative orthant) as discussed in section 3.3.
>
> *References*
>  [1] Shen, Zebang, et al. "An agnostic approach to federated learning with class imbalance." International Conference on Learning Representations. 2021.
>  [2] Hounie, Ignacio, Luiz FO Chamon, and Alejandro Ribeiro. "Automatic data augmentation via invariance-constrained learning." International Conference on Machine Learning. PMLR, 2023.
>
> ## Invariant Learning Results
>
> In Table 4 (Appendix G.3.2) we  compared to Augerino [4], which is a popular invariant learning method. In an easy dataset like MNIST our approach shows similar performance whereas in a more challenging dataset like FMNIST - except for the fully rotated version - our method performs statistically significantly better.
>
> | Dataset | Method        | Fully Rotated             | Partially Rotated         | Translated                 | Scaled                    | Original                  |
> |---------|---------------|---------------------------|---------------------------|----------------------------|---------------------------|---------------------------|
> | MNIST   | Augerino      | $\mathbf{97.78 \pm 0.03}$ | $96.38 \pm 0.00$          | $94.65 \pm 0.01$           | $97.53 \pm 0.00$          | $98.44 \pm 0.00$          |
> | MNIST   | Unconstrained | $94.49 \pm 0.12$          | $96.25 \pm 0.13$          | $94.64 \pm 0.20$           | $97.47 \pm 0.03$          | $98.45 \pm 0.06$          |
> | MNIST   | Constrained   | $94.55 \pm 0.18$          | $96.90 \pm 0.07$          | $93.74 \pm 0.07$           | $97.92 \pm 0.15$          | $98.74 \pm 0.08$          |
> | MNIST   | Resilient     | $95.38 \pm 0.18$          | $\mathbf{97.19 \pm 0.09}$ | $\mathbf{95.21 \pm 0.15}$ | $\mathbf{98.20 \pm 0.04}$ | $\mathbf{98.86 \pm 0.02}$ |
> | F-MNIST | Augerino      | $\mathbf{85.28 \pm 0.54}$ | $81.48 \pm 0.49$          | $81.13 \pm 0.77$           | $83.17 \pm 0.46$          | $90.09 \pm 0.20$          |
> | F-MNIST | Unconstrained | $77.94 \pm 0.06$          | $81.57 \pm 0.36$          | $79.23 \pm 0.17$           | $82.99 \pm 0.18$          | $90.20 \pm 0.23$          |
> | F-MNIST | Constrained   | $84.96 \pm 0.12$          | $85.66 \pm 0.32$          | $83.61 \pm 0.10$           | $86.49 \pm 0.09$          | $91.02 \pm 0.02$          |
> | F-MNIST | Resilient     | $\mathbf{85.57 \pm 0.26}$ | $\mathbf{86.48 \pm 0.15}$ | $\mathbf{85.06 \pm 0.23}$  | $\mathbf{87.26 \pm 0.14}$ | $\mathbf{91.55 \pm 0.31}$ |
>
> *Table 4: Classification accuracy for synthetically invariant datasets. We use the same invariance constraint level $\epsilon_i=0.1$ for all datasets and transformations. We report the mean and standard deviation computed across three independent runs.*
>
>
>  [4] Benton, Gregory, et al. "Learning invariances in neural networks from training data." Advances in neural information processing systems 33 (2020): 17605-17616.

---

> > ### Comment · Reviewer_tedN · 2023-08-19
> > **Thank authors for their response.**
> >
> > I think the authors' responses have addressed all my concerns. I keep my orginal score.

---

### Author Rebuttal · Authors · 2023-08-09

# Response to all Reviewers

We sincerely thank all reviewers for their efforts in reviewing our paper. We are glad to see that all reviewers have expressed that this is a novel and well-motivated method with high potential impact in practical applications. We are also thankful for the reviewers insights and feedback which we find pertinent and valuable. We have carefully addressed your comments and suggestions and hope that the forthcoming exchanges and discussions can lead to further improvements.

The major concern about our work is whether there is sufficient empirical evaluation of resilient constrained learning. We believe that our experiments provide sufficient empirical evaluation of resilient constrained learning. We developed this idea as a way of striking compromises in situations where multiple conflicting requirement result in poor performance. Our experiments show that this does happen in practice. Resilient solutions sacrifice stringent constraints to improve overall performance. For instance, the results shown in Figures 2(left) and 8 (in the supplementary materials) demonstrate that more stringent requirements are relaxed more and Figures 2 (right), 7 and 8 along with Tables 1 and 4 demonstrate that these relaxations lead to performance improvements as measured by the objective loss.

That said, the reviewers' comment that more evidence is required is, as we said above, pertinent and valuable. We have therefore added new results based on the reviewers feedback. In particular, we have performed the following additional experimental analyses:

- *Response Table 1*: We compare average performance of resilient constrained learning with benchmarks for federated learning with class imbalance. It is notable that average performance is comparable with benchmarks.
- *Response Table 2*: We show spread metrics of accuracy -- maximum range and interquantile range -- across clients. Constrained resilient learning reduces the interquantile range at the cost of increasing the maximum range. I.e., it improves the accuracy of most clients at the cost of reducing the accuracy of a few.
- *Response Table 3*: We rank clients according to their realized losses in standard constrained learning and resilient constrained learning. We then compare the relative performance of equally ranked clients. We see that a large fraction of clients have 1% to 5% better accuracy at the cost of possibly substantial decreases in the accuracy of a few outlier clients.
- *Response Tables 4 and 5*: We consider cost functions $h(u) = \| u\|_\beta$ with varying values of $\beta$ in the federated and invariant learning setup.
- *Response table 6*: Ablation on dual and resilient learning rates for the federated learning setup, showing that our method is not overly sensitive to these hyperparameters.

We also highlight in the individual responses several numerical experiments that were already included in the supplementary materials of our submission. The reviewers feedback suggests that having some of these results in the main body of the paper would make for a stronger contribution. We will streamline the presentation of the results in future versions of the manuscript in order to include some of these results in the main body.

We have also included several numerical experiments to highlight properties of resilient constrained learning. For instance, Figures 3 (left), 5 and 6 show decreased sensitivity to problem specification and Figures 3 (right) and 7 show better empirical approximation of the underlying statistical problem. Both of these are properties that resilient cosntrained learning has by definition.

---

> ### Author Response · Authors · 2023-08-10
> **Summary of Results in Rebuttal**
>
> ## Performance (reviewers *4YtT* and *69dA*)
> ### Federated learning with class imbalance
>
> We point out that federated learning with class imbalance, also known as heterogeneous federated learning, is a problem motivated by practical considerations which has received substantial attention; see e.g [R1] and references therein. We adopt it as an example in this paper as a prototypical situation in which accomodating some clients -- i.e., constraints -- can be much more difficult than accomodating most clients.
>
> We adopt as benchmarks Ratio-Loss [R2] and CLIMB [R3]. The latter is a constrained learning approach which we adopt with the modification that constrains cab be relaxed as per the definition of resilient equilibrium.
>
>  Also, we did not tune any hyperparameters and used the same hyperparameters as in [R3], since the objective was to provide a fair comparison. In order to see how the hyperparameters that are exclusive to our method, namely $\eta_u$ and $h(u)$, we conducted ablations detailed below.
>
> We also thank reviewers *4YtT* and *69dA* for their suggestion to include more quantitative metrics. In this setup, average accuracy is not really the pertinent metric. Observe that the *average* accuracy (Response Table 1) is  overall similar to the constrained approach, and consistently higher than the baseline [R3].
>
> | Dataset | Imb. Ratio | Num Minority | Ratio-Loss[R2] | CLIMB[R3]  | OURS   |
> |---------|------------|--------------|------------|--------|--------|
> | F-MNIST | 10         | 3            | $92.5$     | $92.6$ | $93.4$ |
> | F-MNIST | 20         | 3            | $94.0$     | $93.8$ | $94.4$ |
> | CIFAR10 | 10         | 3            | $81.3$     | $81.5$ | $81.5$ |
> | CIFAR10 | 20         | 3            | $83.4$     | $82.4$ | $82.6$ |
>
> **Response Table 1**: Average  Accuracy for different setups. The imbalance ratio denotes the fraction of samples kept in minority classes. As in [R4] we use a dirichlet distribution to allocate samples among clients, with parameter $d=0.3$.
>
> However, the distribution o performance among clients varies. We actually chose this experiment as one in which the differences between resilient learning and standard constrained learning are apparent. Resilient learning has (generally) less spread in the interquartile range and higher maximum spread (Response Table 2) than the constrained approach.
>
> | Dataset | Imb. Ratio | Ratio Loss[R2]     | CLIMB[R3]          | OURS           |
> |---------|------------|----------------|----------------|----------------|
> | F-MNIST | 0.1        | $2.2$ $(64.1)$ | $3.5$ $(28.6)$ | $2.4$ $(50.6)$ |
> | F-MNIST | 0.05       | $2.2$ $(50.2)$ | $2.7$ $(28.6)$ | $1.7$ $(45.8)$ |
> | CIFAR10 | 0.1        | $8.1$ $(49.4)$ | $8.7$ $(35.8)$ | $8.7$ $(46.4)$ |
> | CIFAR10 | 0.05       | $8.6$ $(44.5)$ | $8.4$ $(32.0)$ | $7.9$ $(41.1)$ |
>
> **Response Table 2**: Client accuracy spread metrics for different setups. The first number denotes interquantile range and the number in parentheses denotes the maximum minus the minimum accuracy, both computed across 100 clients.
>
> This is precisely what the method is designed to do. Sacrifice the performance of outliers to benefit the performance of the majority of the agents. In order to showcase this, we order clients by their accuracy. We then compute the fraction of clients in the resilient method that outperform equally ranked clients for baseline methods (Response Table 3).
> | Dataset | Imb. Ratio | Improved (%) | Mean Improvement | Max Improvement | Mean Decrease | Max Decrease |
> |---------|------------|--------------|------------------|-----------------|---------------|--------------|
> | CIFAR10 | 10         | 77           | 0.4              | 1.5             | 0.5           | 10.0         |
> | CIFAR10 | 20         | 79           | 0.5              | 2.1             | 0.3           | 9.1          |
> | F-MNIST | 10         | 92           | 1.6              | 4.8             | 0.9           | 23.3         |
> | F-MNIST | 20         | 94           | 1.3              | 2.6             | 0.7           | 19.4         |
>
> **Response Table 3**: Changes in accuracy for equally ranked clients for the resilient method.
> As intended, performance improves for most clients, though at the cost of a decrease in performance for a few *outliers*.
>
> *References*
>
> [R1] Zhu, Hangyu, et al. "Federated learning on non-IID data: A survey." Neurocomputing 465 (2021): 371-390.
>  [R2] Wang, Lixu, et al. "Addressing class imbalance in federated learning." Proceedings of the AAAI Conference on Artificial Intelligence. Vol. 35. No. 11. 2021.
>  [R3] Shen, Zebang, et al. "An agnostic approach to federated learning with class imbalance." International Conference on Learning Representations. 2021.
>  [R4] Durmus, Alp Emre, et al. "Federated Learning Based on Dynamic Regularization." International Conference on Learning Representations. 2021.

---

> > ### Author Response · Authors · 2023-08-10
> > **Summary of Results in Rebuttal**
> >
> > ## New Ablations  (as suggested by reviewer 4YtT)
> >
> > ### Choice of cost function
> >  We run $||u||_\beta$ with $\beta =1, 2, 4$, and infinity, for fashion MNIST in both the federated and invariant setting. As discussed in Section 3.3, $\beta = 1$ recovers penalty based methods.
> > **Federated Learning**
> > | $\beta$    | Mean Acc | IQR   | Max Range |
> > |------------|----------|-------|-----------|
> > | 1.0        | $93.3$   | $2.5$ | $49.5$    |
> > | 2.0        | $93.4$   | $2.4$ | $50.6$    |
> > | 4.0        | $93.4$   | $2.6$ | $45.6$    |
> > | $\infty$   | $92.7$   | $3.8$ | $28.1$    |
> >
> > **Response Table 4**:*Cost function ablation for the heterogeneus federated learning in fashion-MNIST using 100 clients, 3 minority classes, an imbalance ratio of $\rho=10$ and dirichlet allocation with parameter $d=0.3$. We report the mean, interquartile range and range (maximum value minus minimum value) test accuracy across clients.*
> >
> > **Invariance**
> >
> >  | $\beta$  | Partially Rotated | Translated      | Scaled          |
> > |----------|-------------------|-----------------|-----------------|
> > | 1.0      | $86.08\pm 0.38$   | $86.85\pm 0.20$ | $85.02\pm 0.46$ |
> > | 2.0      | $85.37\pm 0.17$   | $86.65\pm 0.25$ | $84.92\pm 0.39$ |
> > | 4.0      | $85.23\pm 0.20$   | $86.64\pm 0.11$ | $84.65\pm 0.68$ |
> > | $\infty$ | $82.94\pm 0.17$   | $85.47\pm 0.19$ | $83.35\pm 0.49$ |
> >
> > **Response Table 5**:*Cost function ablation for invariant fashion-MNIST datasets. We compute the mean and standard deviation in test accuracy across three independent runs.*
> >
> > ### Learning Rates
> > We run an ablation on $\eta_u, \eta_\lambda$, the perturbation and dual learning rates over a small grid of 12 values, and find that, in this range, the performance of the algorithm is not overly sensitive to this choice. We also observe that  that the rates that were used in the paper ($\eta_u = 0.1, \eta_\lambda = 2$) are not optimal in this setup, and thus further improvements in performance could be obtained through more extensive hyperparameter tuning, but this was not the focus of our experiments.
> >
> > | $\eta_u$ \ $\eta_\lambda$ | 0.1  | 0.5  | 1    | 2    |
> > |-----------------------|------|------|------|------|
> > | 0.1                   | 81.4 | 81.6 | 81.8  | 81.8 |
> > | 0.5                   | 81.4 | 80.9 | 81.4 | 81.1 |
> > | 1                     | 81.6 | 81.6 | 81.6 | 81.7 |
> >
> > **Response Table 6**:*Dual and resilient learning rate ablation in Heterogenous federated learning setting. We report mean Test Accuracy for CIFAR100 using 100 clients, 3 minority classes, an imbalance ratio of $\rho=10$ and dirichlet allocation with parameter $d=0.3$.*
> >
> > ## Comparisons to constrained learning and discussion (reviewers *69dA*, *iyNP* and *4YtT*)
> >
> > In addition to the quantitative metrics presented above, the results included both in the experimental section and appendix mainly aim to highlight the differences between our approach and existing constrained learning approaches (reviewers *69dA* and *iyNP*) . In doing so, we discuss the inherent trade-offs or limitations of out approach (reviewers *69dA* and *4YtT*). Here we provide a brief summary and expand on how these address the fundamental aspects of our method.
> >
> > **Constraint Relaxation and Relative difficulty**:
> > This illustrates how our method adapts $u$ depending on how difficult it is to satisfy the constraint, whereas the constrained approach enforces the constraint ($u=0$) regardless of the price to pay in performance. Since the impact on performance depends on the data distribution, the most imbalanced clients (Figure 2 in main text) are relaxed more. This shows empirically that our method behaves as intended *in a learning setup*. The same happens for invariances that are not present in the dataset (Figure 8 in Appendix).
> >
> > **Controlling the performance  vs. relaxation trade-off**:
> > These experiments (Figure 2 in main text, Figure 4 in the Appendix) illustrate the inherent trade-off between imposing constraints and having better performance in terms of statistical risk in the objective.

---

### Comment · Area_Chair_oE63 · 2023-08-18
**To Reviewers: Please respond to the author rebuttals.**

Dear reviewers,

Thank you for serving as a reviewer for NeurIPS!

We are towards the end of the discussion stage with authors, but some of you haven't posted your response to the author rebuttals yet.
As the scores for this paper are diverse, please check the author rebuttals, reply to them and update your score (when necessary) ASAP. Thanks!

-AC

---

### Decision · Program_Chairs · 2023-09-21

**Decision:**

Accept (poster)

**Comment:**

This paper introduces Resilient Constrained Learning, a novel approach that strikes a balance between optimizing the primary objective and relaxing constraints. The contributions, such as sensitivity-based relaxation and a practical algorithm, have garnered positive reviews from the reviewers. Notably, the paper excels in terms of clarity, adaptability to real-world requirements, and its potential for significant impact. Both theorems and experiments provide substantial evidence of its effectiveness. In conclusion, the meta-reviewer finds the contribution of this paper to be worthy of publication and recommends that the authors incorporate the feedback received during the preparation of the camera-ready version.